EMBO
Molecular Medicine

# Inhibition of Stat3-mediated astrogliosis ameliorates pathology in an Alzheimer's disease model

Nicole Reichenbach[1], Andrea Delekate[1], Monika Plescher[1], Franziska Schmitt[1], Sybille Krauss[1], Nelli Blank[1], Annett Halle[1,2] & Gabor C Petzold[1,3,*] (iD)

## Abstract

Reactive astrogliosis is a hallmark of Alzheimer's disease (AD), but its role for disease initiation and progression has remained incompletely understood. We here show that the transcription factor Stat3 (signal transducer and activator of transcription 3), a canonical inducer of astrogliosis, is activated in an AD mouse model and human AD. Therefore, using a conditional knockout approach, we deleted Stat3 specifically in astrocytes in the APP/PS1 model of AD. We found that Stat3-deficient APP/PS1 mice show decreased β-amyloid levels and plaque burden. Plaque-close microglia displayed a more complex morphology, internalized more β-amyloid, and upregulated amyloid clearance pathways in Stat3-deficient mice. Moreover, astrocyte-specific Stat3-deficient APP/PS1 mice showed decreased pro-inflammatory cytokine activation and lower dystrophic neurite burden, and were largely protected from cerebral network imbalance. Finally, Stat3 deletion in astrocytes also strongly ameliorated spatial learning and memory decline in APP/PS1 mice. Importantly, these protective effects on network dysfunction and cognition were recapitulated in APP/PS1 mice systemically treated with a preclinical Stat3 inhibitor drug. In summary, our data implicate Stat3-mediated astrogliosis as an important therapeutic target in AD.

**Keywords** Alzheimer's disease; astrocytes; astrogliosis; glia; Stat3
**Subject Categories** Immunology; Neuroscience; Pharmacology & Drug Discovery

## Introduction

Alzheimer's disease (AD) is a common and chronic neurodegenerative disorder characterized by cognitive decline. An early and characteristic hallmark of the disease in patients and mouse models is that astrocytes around β-amyloid (Aβ) plaques undergo prominent morphological changes, known as reactive astrogliosis (Carter *et al*, 2012; Pekny *et al*, 2015). While reactive astrogliosis is a ubiquitous and sustained response mechanism of the brain to acute or chronic injury (Burda & Sofroniew, 2014; Pekny & Pekna, 2014), its underlying mechanisms and the consequences for AD pathogenesis and other conditions have remained incompletely understood. However, given that the primary and most important role of astrocytes in the brain is to maintain neuronal health (Barres, 2008), it seems intuitive that astrogliosis may be an important target to alleviate neurodegeneration in brain diseases. Previous studies have shown that pharmacological or transgenic interference with astroglial signaling pathways that are pathologically activated in models of chronic neurodegenerative diseases including AD can restore network function and support neuronal survival (Delekate *et al*, 2014; Tong *et al*, 2014; Reichenbach *et al*, 2018). However, a preclinical exploration of reactive astrogliosis as a potential therapeutic target in AD has been lacking.

One of the key regulators of many pathways associated with astrogliosis is signal transducer and activator of transcription 3 (Stat3), a transcription factor that is activated through phosphorylation by Janus kinases (JAK) in response to cytokines, intercellular mediators, and growth factors (Villarino *et al*, 2017). JAK/Stat3 activation in astrocytes has been detected in a plethora of conditions and disease models that are associated with astrogliosis, including stroke (Justicia *et al*, 2000), AD (Ben Haim *et al*, 2015), spinal cord injury (Okada *et al*, 2006; Herrmann *et al*, 2008), multiple sclerosis (Qin *et al*, 2012), and epilepsy (Xu *et al*, 2011). Interestingly, but perhaps not surprisingly given that different context-dependent variations of astrogliosis exist (Liddelow *et al*, 2017), the consequences of Stat3 activation in astrocytes appear to depend on the severity of astrogliosis as well as the underlying disease. For example, inhibiting Stat3 activation in astrocytes in spinal cord injury, where strong astrogliosis contributes to scar formation, worsens outcome (Okada *et al*, 2006; Herrmann *et al*, 2008), but has the opposite effect in models of multiple sclerosis (Qin *et al*, 2012) or neonatal hypoxic brain damage (Hristova *et al*, 2016). Hence, Stat3 may aggravate or alleviate neurodegeneration in a context-dependent manner (Tyzack *et al*, 2017), indicating that the full repertoire of cellular pathways initiated by Stat3 signaling remains to be investigated.

1  German Center for Neurodegenerative Diseases (DZNE), Bonn, Germany
2  Department of Neuropathology, University Hospital Bonn, Bonn, Germany
3  Department of Neurology, University Hospital Bonn, Bonn, Germany
   *Corresponding author. Tel: +49 228 43302684; Fax: +49 228 43302683; E-mail: gabor.petzold@dzne.de

We aimed to preclinically test the applicability of Stat3-mediated astrogliosis as a potential target for future therapies. Therefore, we sought to determine the molecular pathways and consequences of an astrocyte-specific conditional deletion of Stat3 on pathological hallmarks, network function, neuroinflammation, and cognitive dysfunction in an AD mouse model.

# Results

## Effective astrocyte-specific deletion of Stat3, a regulator of astrogliosis in AD

To delete Stat3 in astrocytes, we used mice expressing tamoxifen-inducible Cre recombinase (CreERT) under the astrocyte-specific connexin-43 (Cx43) promoter (Eckardt *et al*, 2004). To confirm efficacy and astrocyte selectivity, we first crossed these mice to the Ai9 (tdTomato-loxP) reporter line that expresses tdTomato following Cre recombination. Injection of tamoxifen into 6-week-old Cx43-Cre::tdTomato-loxP mice resulted in strong tdTomato expression. Immunohistochemistry at 8 or 11 months revealed a high selectivity for astrocytes. ~90 % of tdTomato-positive cells were astrocytes (Fig 1A and B), and the small fractions of remaining cells were either neurons (~5–6%), NG2 cells (~4%), or oligodendrocytes (< 1%) (Fig 1A and B).

Next, we investigated the role of Stat3 in the APPswe/PS1ΔE9 (APP/PS1) mouse model of AD (Jankowsky *et al*, 2004), using a mouse line in which exons 12–14 of the Stat3 gene are flanked by loxP sites, enabling Cre-mediated functional Stat3 deletion (Alonzi *et al*, 2001). We then generated APP/PS1$^{tg/wt}$::Cx43-Cre$^{tg/wt}$::Stat3$^{loxP/loxP}$ (APP/PS1-Stat3KO) mice and activated Cre recombination by tamoxifen injection at 6 weeks. APP/PS1$^{tg/wt}$::Cx43-Cre$^{wt/wt}$::Stat3$^{loxP/loxP}$ (APP/PS1-Stat3WT), APP/PS1$^{wt/wt}$::Cx43-Cre$^{tg/wt}$::Stat3$^{loxP/loxP}$ (WT-Stat3KO), and APP/PS1$^{wt/wt}$::Cx43-Cre$^{wt/wt}$::Stat3$^{loxP/loxP}$ (WT-Stat3WT) mice served as controls and were also injected with tamoxifen at the same age. We found that 46 ± 10% of all GFAP-positive astrocytes in the cortex and 41 ± 8% in the hippocampus were positive for pStat3. These ratios were even higher in the peri-plaque regions, where the majority of reactive astrocytes in APP/PS1-Stat3WT mice were positive for activated Stat3 (cortex, 69 ± 11%; hippocampus, 77 ± 14%; Fig 1C and D), confirming earlier reports (Ben Haim *et al*, 2015), whereas Stat3 activation in WT-Stat3WT was negligible (Fig 1C). Importantly, Stat3 activation in astrocytes was reduced by ~80% in APP/PS1-Stat3KO in the cortex and hippocampus (Fig 1C–E), confirming strong and astrocyte-selective Stat3 deletion in these mice.

Importantly, we also detected nuclear pStat3 activation in the majority of reactive peri-plaque astrocytes in postmortem cortical or hippocampal samples from AD patients (Fig 1F and G).

## Stat3 regulates plaque-associated astrocytic and microglial complexity

Next, we determined the structural and functional consequences of the astrocyte-specific Stat3 deletion in APP/PS1 mice. Overall astroglial and microglial coverage remained mostly unchanged in APP/PS1-Stat3KO compared to APP/PS1-Stat3WT mice (Fig 2A and B). Moreover, we did not detect markers of astroglial proliferation

around plaques in APP/PS1-Stat3KO or APP/PS1-Stat3WT mice, similar to previous reports (Wang *et al*, 2018) that amyloid pathology does not induce astrocyte proliferation. The relative density of microglia and astrocytes in both lines was also similar (Fig EV1). Importantly, however, we detected significant morphological alterations in the peri-plaque region. Specifically, we found a volume increase in astrocytes near Aβ plaques, which was paralleled by longer astroglial processes (Fig 2C and D). Interestingly, however, these changes were relatively minor compared to peri-plaque microglia, which displayed a higher number of branches and process junctions as well as longer processes (Fig 2E and F). Some of these changes were also evident in plaque-distant astrocytes and microglia, although they were less pronounced and did not reach statistical significance (Fig EV2).

## Astrocyte-selective Stat3 deletion results in reduced plaque load and dystrophic neurites by modulating Aβ internalization and clearance

To investigate the consequences of this higher astroglial and microglial peri-plaque complexity, we next quantified Aβ levels and plaque burden in APP/PS1-Stat3KO compared to APP/PS1-Stat3WT mice. Interestingly, we found that plaque load was strongly reduced in APP/PS1-Stat3KO mice (Fig 3A). This effect was caused by a smaller size of plaques, but not by decreased plaque density (Fig 3B and C). Moreover, the reduction in plaque burden induced by Stat3 deletion in astrocytes also led to a decrease in soluble Aβ$_{1-40}$ and Aβ$_{1-42}$ in the brain (Fig 3D and E). The expression of amyloid precursor protein (APP) and its C-terminal degradation fragments remained unchanged (Fig 3F–H), suggesting that the effects of astroglial Stat3 signaling were mediated by increased Aβ clearance and not altered APP expression or processing. To further explore this notion, we performed an immunohistochemical engulfment/phagocytosis assay (Schafer *et al*, 2012) based on high-resolution stacks of mouse brain sections stained for astrocytes, microglia, and Aβ (Fig 4A). These analyses revealed that the astrocyte-specific Stat3 deletion in APP/PS1 mice led to a significant enhancement of fibrillar Aβ in microglia, as quantified by IC16 antibody and methoxy-XO4 co-localization with microglia, respectively (Fig 4B and C; Movie EV1). In contrast, we detected very little Aβ in astrocytes in comparison with microglia, and found no effect of Stat3 (Fig 4B and C; Movie EV2), indicating that Stat3-mediated astrogliosis stimulates Aβ clearance by microglia. In line with this interpretation, expression of the microglial Aβ-degrading enzymes neprilysin/CD10 as well as CD36 was increased in APP/PS1-Stat3KO mice (Fig 4D, E and H). Moreover, as Aβ aggregation and plaque deposition are promoted by the binding of Aβ to apolipoprotein E (apoE) in mouse models and humans (Verghese *et al*, 2011), we also determined mouse apoE levels in our model and found a significant reduction in apoE in APP/PS1-Stat3KO compared to APP/PS1-Stat3WT mice (Fig 4F and H). In contrast, expression of the triggering receptor expressed on myeloid cells 2 (TREM2) remained unchanged (Fig 4G and H).

## Astrocyte Stat3 deletion induces a phenotypical switch in astrocytes

Astrocytes in AD models express transcripts associated with a potentially neurotoxic "A1" phenotype, as opposed to a potentially

**Figure 1. Stat3 activation in an AD model is efficiently reduced using a conditional knockout strategy.**

A, B   To verify the specificity of Cx43-CreERT mice for astrocytes, mice were crossed to a tdTomato reporter line to induce Cre-dependent tdTomato expression after tamoxifen administration. TdTomato fluorescence was enhanced with an antibody against red fluorescent protein, and astrocytes were stained with antibodies against GFAP or S100β (not shown). Merged images and quantification show efficient and widespread tdTomato expression specific for astrocytes (arrowheads indicate tdTomato and GFAP overlap) in 8- and 11-month-old animals. Expression was low in neurons (stained with NeuN, not shown) and negligible in NG2 cells and oligodendrocytes. Scale bars, 250 μm (upper panel) and 50 μm (lower panel).

C   The majority of peri-plaque astrocytes in APP/PS1-Stat3WT were positive for activated/phosphorylated Stat3 (pStat3) in the cortex (Cx) and hippocampus (HC), whereas the fraction of pStat3-positive astrocytes was significantly lower in APP/PS1-Stat3KO mice in both regions (*n* = 8 mice (three females and five males) per group; age, 8 months; *P* < 0.05, Mann–Whitney test). The occurrence of pStat3 in WT-Stat3KO and WT-Stat3WT mice was negligible (*n* = 8 mice (four females and four males) per group; age, 8 months; Mann–Whitney test).

D, E   Examples of Stat3 activation in APP/PS1-Stat3KO and APP/PS1-Stat3WT. (D) pStat3 (arrowheads) was abundantly present in reactive astrocytes (identified by GFAP) around plaques (identified by IC16). (E) Only few astrocytes were positive for pStat3 in APP/PS1-Stat3WT mice. Scale bars, 50 μm.

F, G   In postmortem brain sections from AD patients, we detected nuclear pStat3 (arrows) in the majority of peri-plaque reactive astrocytes (identified by GFAP) in the cortex and hippocampus (plaques were stained with methoxy-XO4, arrowheads). Scale bar, 50 μm. In total, *n* = 4 cortical and *n* = 3 hippocampal sections were analyzed.

Data information: Data are represented as mean ± SEM.

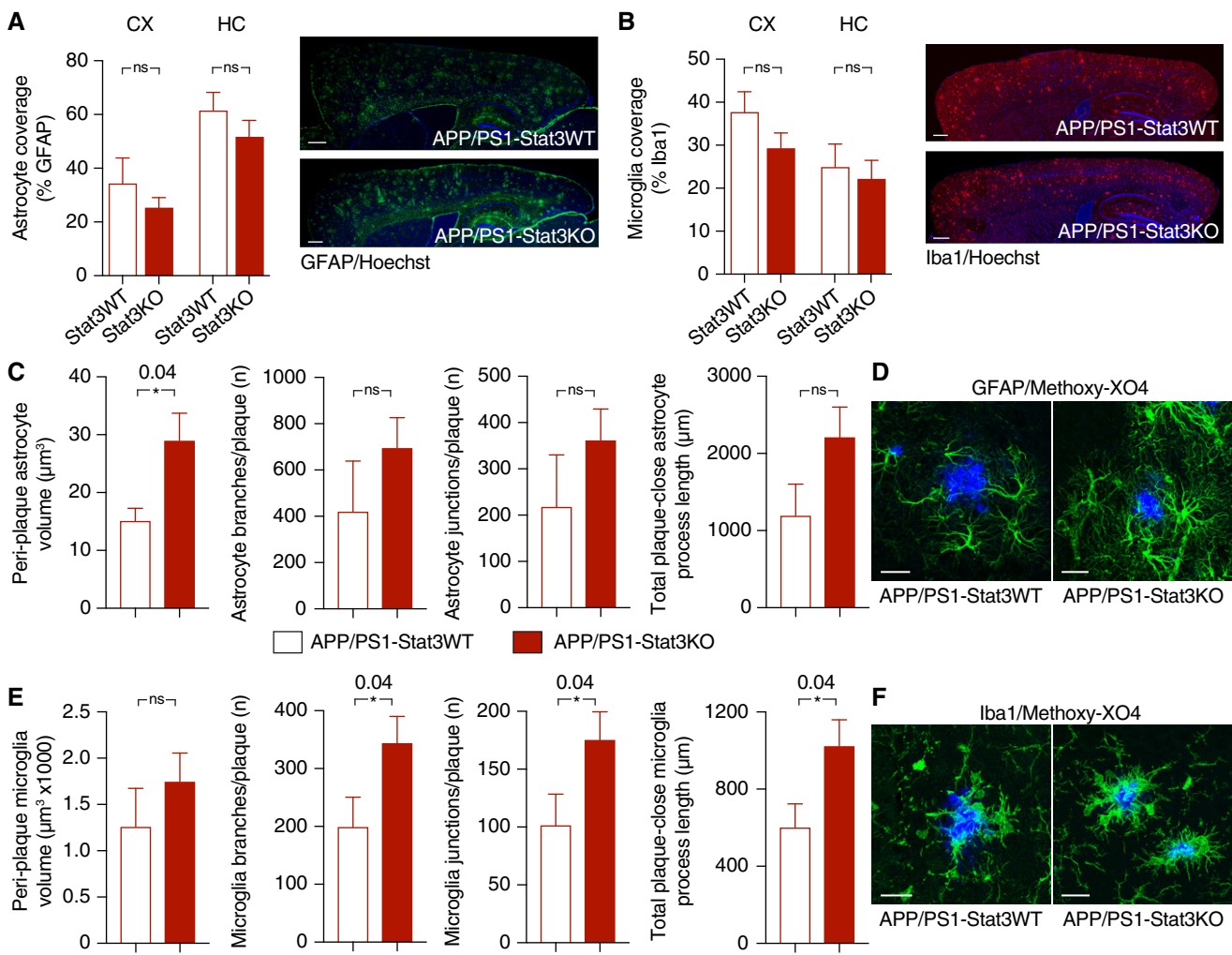

**Figure 2. Astrocyte-specific Stat3 deletion induces greater astrocytic and microglial complexity.**

A, B    Overall astrocytic and microglial coverage, as assessed by GFAP and Iba1 stainings, remained unchanged in APP/PS1-Stat3WT versus APP/PS1-Stat3KO mice (n = 8 mice (four females and four males) per group; age, 8 months; Mann–Whitney test test for each comparison). Scale bars, 500 μm.

C, D    The volume of peri-plaque reactive astrocytes was increased by the Stat3 deletion (*P < 0.05, Mann–Whitney test), and there was a nonsignificant trend toward more astrocytic branches and junctions and longer total process length (Mann–Whitney test; APP/PS1-Stat3WT, n = 5 (two females and three males) mice; APP/PS1-Stat3KO, n = 7 (three females and four males) mice; age, 8–10 months). Scale bar, 20 μm.

E, F    Microglial volume was not affected by the Stat3 deletion (Mann–Whitney test), but peri-plaque microglia had significantly more microglial branches and junctions per plaque, and the total process length of peri-plaque microglia was increased (*P < 0.05, Mann–Whitney test; same mice as in C and D). Scale bar, 20 μm.

Data information: Data are represented as mean ± SEM.

neuroprotective "A2" phenotype observed in other conditions (Liddelow *et al*, 2017). Given the enhanced microglial phagocytic capacity in APP/PS1-Stat3KO mice in our study, we next determined the expression of selected "A1" and "A2" transcripts by quantitative PCR in cortex. These experiments showed that the expression of the "A1" markers *Amigo2* and *C3* was significantly downregulated (Fig 5A). In turn, the "A2" marker *Tm4sf1* was upregulated (Fig 5B). As confirmation, Western blot analysis as well as immunohistochemistry against the "A1" marker C3d showed significantly reduced expression in peri-plaque reactive astrocytes in APP/PS1-Stat3KO mice (Fig 5C–F), together indicating an "A1"-to-"A2" switch in astrocytes induced by Stat3 deletion. In line with this, the whole-brain levels of the pro-inflammatory

cytokines IL-1β and TNF-α, which have both been associated with the progression of Aβ deposition, neurodegeneration, and cognitive decline in AD (Heneka *et al*, 2015), were reduced in APP/PS1-Stat3KO compared to APP/PS1-Stat3WT mice (Fig 5G). However, no changes were seen in the levels of IL-10 (Fig 5G), which acts upstream of the JAK/Stat3 pathway (Villarino *et al*, 2017) and is therefore unlikely to be influenced by an astrocyte-specific Stat3 deletion.

We also determined the consequences of enhanced microglial phagocytic capacity on the burden of dystrophic neurites around Aβ plaques, using LAMP1 immunostaining as a proxy marker (Condello *et al*, 2015), and found that the area of dystrophic neurites was reduced in APP/PS1-Stat3KO mice (Fig 5H).

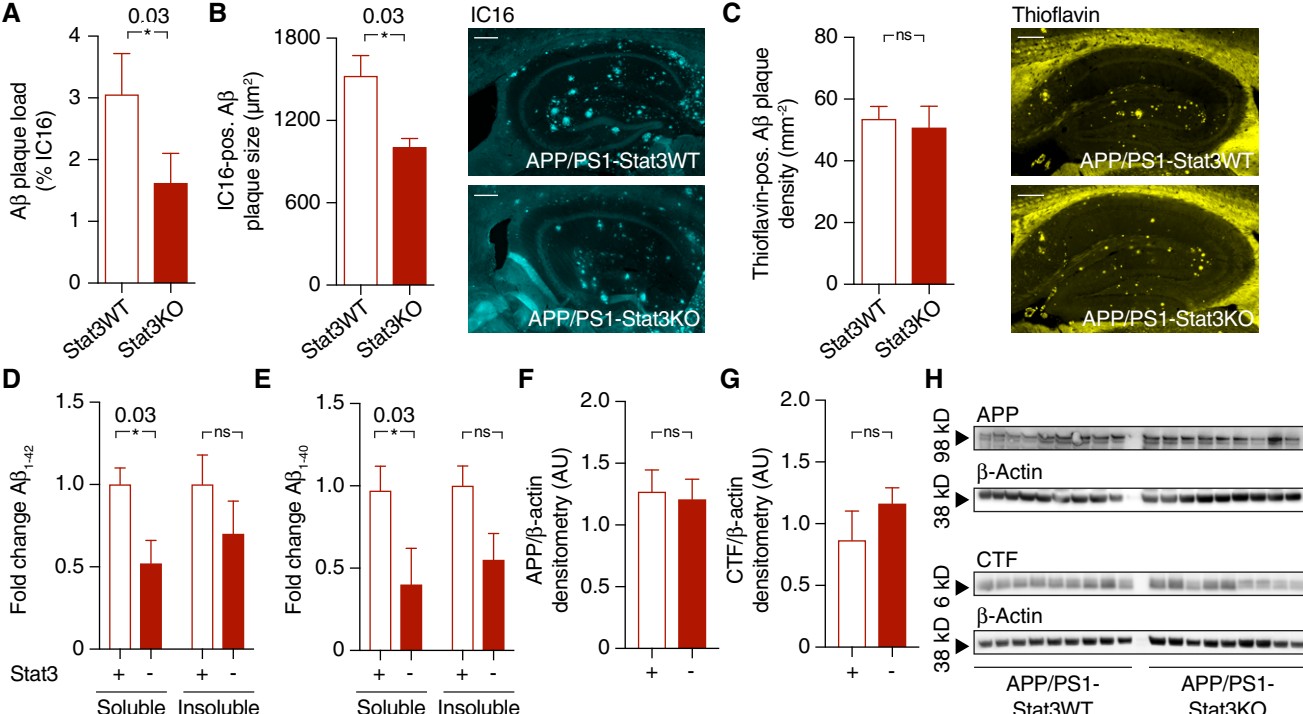

**Figure 3.  Astrocyte-specific Stat3 deletion reduces Aβ plaque burden and Aβ levels without altering APP metabolism.**

A, B    Aβ plaque burden, as assessed by plaque load and size using an anti-Aβ antibody, was strongly reduced in APP/PS1-Stat3KO versus APP/PS1-Stat3WT mice
        (*$P < 0.05$, Mann–Whitney test; scale bar, 300 μm).
C       Plaque density, assessed by staining brain sections with thioflavin (yellow), remained unchanged in APP/PS1-Stat3KO versus APP/PS1-Stat3WT mice (Mann–
        Whitney test; scale bar, 250 μm).
D, E    Electrochemiluminescence ELISA after sequential extraction from whole-brain homogenates using RIPA and SDS buffer revealed that soluble $Aβ_{1-42}$ and $Aβ_{1-40}$
        were significantly reduced (*$P < 0.05$, Mann–Whitney test), whereas there was a nonsignificant trend toward reduced levels of insoluble $Aβ_{1-42}$ and $Aβ_{1-40}$
        (Mann–Whitney test).
F–H     Western blot quantification of full-length amyloid precursor protein (APP) and its C-terminal fragments (CTF) showed no differences between both groups
        (Mann–Whitney test).

Data information: Data are represented as mean ± SEM. For all datasets, $n = 9$ male mice for both groups (age, 11 months).
Source data are available online for this figure.

## Stat3 deletion in astrocytes normalizes cerebral network activity, and a reduction in hyperactivity attenuates Stat3 activation

Aberrant glial–neuronal network activity is a therapeutically relevant hallmark of AD that is strongly linked to cognitive function in mouse models and patients (Palop & Mucke, 2016; Reichenbach *et al*, 2018). Therefore, we determined spontaneous cellular activity of cortical astrocytes and neurons *in vivo* using two-photon microscopy of the calcium indicator OGB-1 in anesthetized mice. Astrocytes were identified by sulforhodamine 101 co-labeling, and Aβ plaques were labeled with the intravital dye methoxy-XO4 (Fig 6A). Interestingly, we found that hyperactivity of astrocytes, which is an important component of network dysregulation in mouse models (Delekate *et al*, 2014; Reichenbach *et al*, 2018), was reduced to wild-type levels in APP/PS1-Stat3KO mice (Fig 6B). Moreover, neuronal activity in the cortex was also reduced to wild-type levels in these mice (Fig 6C and D). These data confirm and extend previous studies showing that reactive astrocytes are an important and potentially detrimental contributing factor to network dysregulation in the brain (Ortinski *et al*, 2010; Reichenbach *et al*, 2018).

However, we also considered a reciprocal scenario in which network dysregulation is not only a downstream effect, but also a cause of reactive astrogliosis, for example, by calcium-dependent signaling pathways (Kanemaru *et al*, 2013). Having previously shown that blockade of the P2Y1 purinoreceptor (P2Y1R) normalizes astroglial hyperactivity in AD mouse models (Reichenbach *et al*, 2018), we therefore determined the effects of chronic P2Y1R inhibition on Stat3 activation. To this end, we applied the P2Y1R inhibitor MRS2179 intracerebroventricularly to APP/PS1-Stat3WT mice continuously for 6 weeks (Fig 6E). As expected (Reichenbach *et al*, 2018), this treatment led to a reduction in astroglial hyperactivity (Fig 6F) and calcium waves (Fig 6G). Interestingly, the number of reactive astrocytes positive for activated Stat3 was also strongly reduced by the treatment (Fig 6H and I).

## Stat3 deletion in reactive astrocytes protects against cognitive decline in APP/PS1 mice

Having found that astroglial Stat3 signaling strongly modulates the inflammatory environment around plaques, Aβ burden and

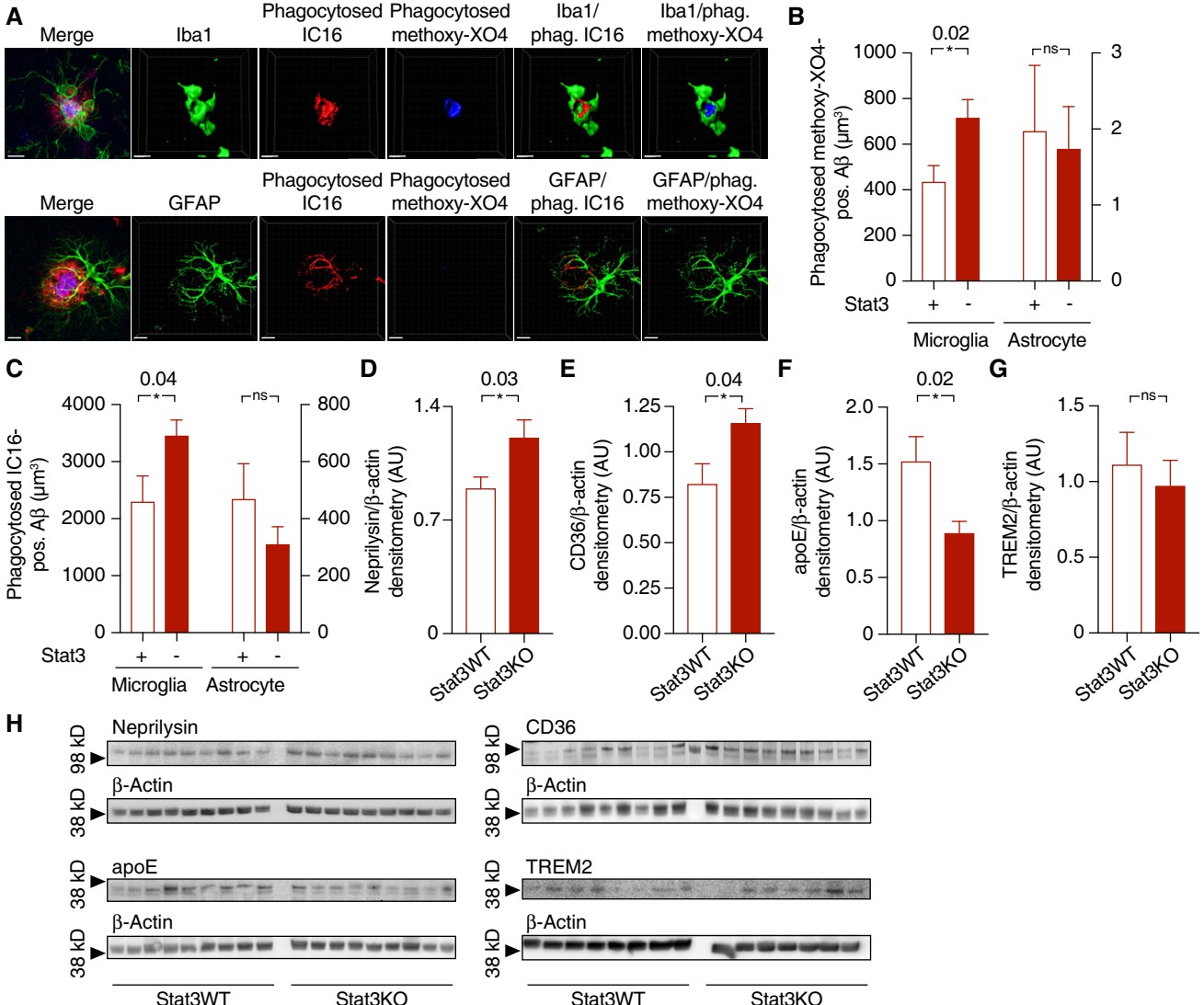

**Figure 4. Astrocyte-specific Stat3 deletion increases microglial Aβ internalization and degradation, and reduces apoE expression, dystrophic neurites, and detrimental cytokines.**

A    Internalization of Aβ (stained with IC16 antibody or methoxy-XO4) was assessed using an engulfment assay, in which glial and Aβ structures were surface-rendered and Aβ volumes co-localized with glial volumes were quantified. Scale bars, 10 μm.

B, C    Microglia (left Y axes) from APP/PS1 mice internalized significantly more Aβ positive for IC16 or methoxy-XO4 when Stat3 was deleted in astrocytes (*$P < 0.05$, Mann–Whitney test), whereas no changes were seen in astrocytes (right axes; APP/PS1-Stat3WT, $n = 8$ (four females and four males) mice; APP/PS1-Stat3KO, $n = 11$ (five females and six males) mice; age, 11 months; Mann–Whitney test).

D–H    (D–F) Western blot quantification of protein levels of the Aβ-degrading enzymes neprilysin/CD10 and CD36, as well as the Aβ-binding apolipoprotein E (apoE), revealed a significantly increased expression of neprilysin and CD36 and a decreased expression of apoE (APP/PS1-Stat3WT, $n = 9$ (five females and four males) mice; APP/PS1-Stat3KO, $n = 9$ (five females and four males) mice; age, 11 months; *$P < 0.05$, Mann–Whitney test for all comparisons). (G) In contrast, TREM2 expression remained unchanged (APP/PS1-Stat3WT, $n = 8$ (four females and four males) mice; APP/PS1-Stat3KO, $n = 7$ (four females and three males) mice; age, 11 months; Mann–Whitney test). (H) Western blots for proteins analyzed in (D-G).

Data information: Data are represented as mean ± SEM.
Source data are available online for this figure.

clearance, neurite degeneration, and network function, we next determined its effects on the severity and progression of cognitive decline. To this end, we subjected APP/PS1-Stat3KO, APP/PS1-Stat3WT, WT-Stat3KO, and WT-Stat3WT mice to the Morris Water Maze paradigm of spatial learning and memory. As expected, APP/PS1-Stat3WT showed a significantly longer latency to reach the hidden platform during the training phase (Fig 7A and B). Interestingly, however, APP/PS1-Stat3KO showed a similar learning curve as WT-Stat3KO and WT-Stat3WT mice (Fig 7A and B), indicating a strong positive effect of astroglial Stat3 deletion on spatial learning. The swimming velocity was similar in all groups, indicating no overt motor phenotype associated with the cell-specific deletion

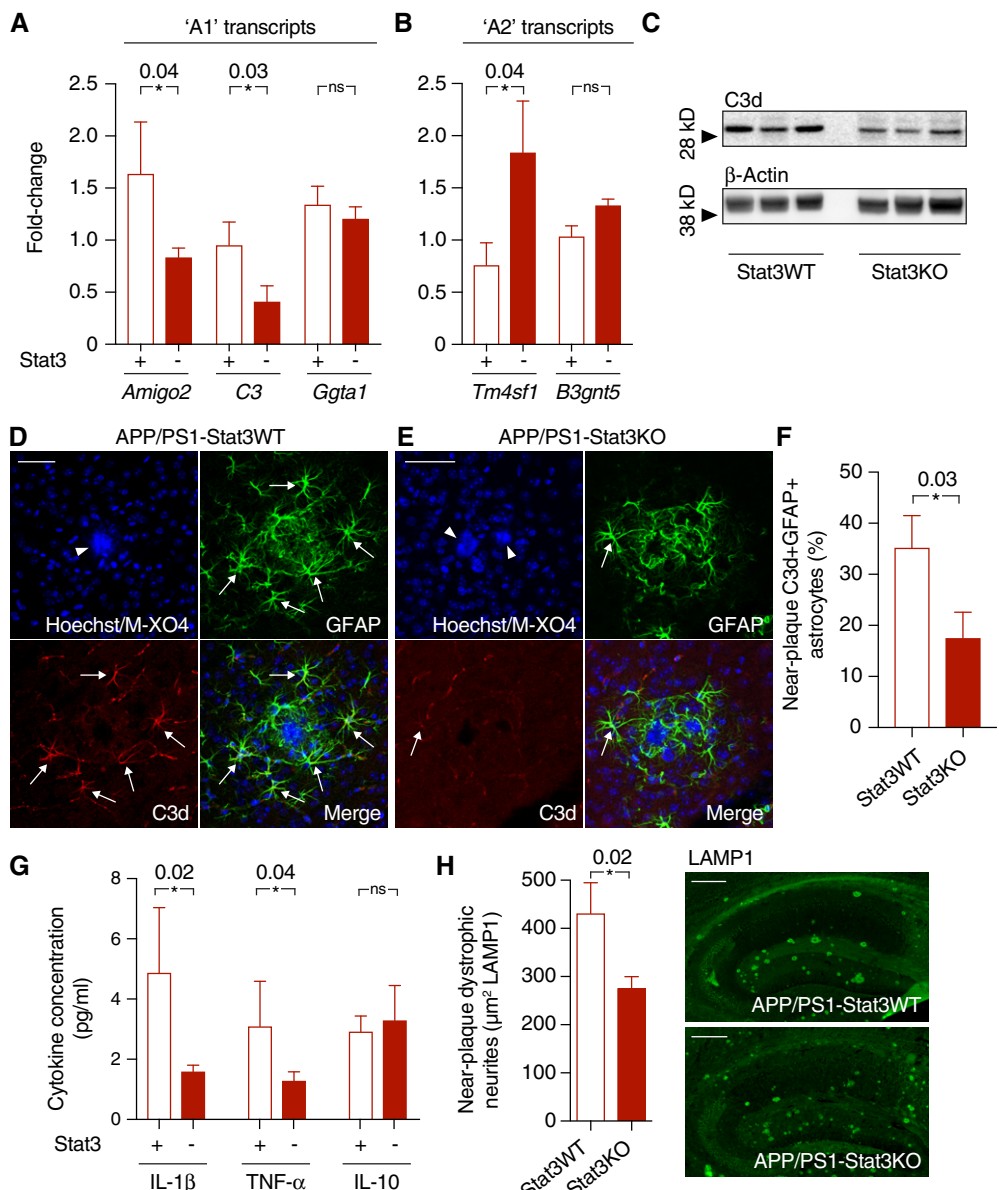

**Figure 5. Stat3 deletion triggers a phenotypical switch in reactive astrocytes.**

A, B   Quantitative PCR from cortex of APP/PS1-Stat3KO compared to APP/PS1-Stat3WT mice revealed lower expression of "A1" markers *Amigo2* and *C3*, whereas *Ggta1* remained unchanged. In turn, the "A2" marker *Tm4sf1* was upregulated and there was a nonsignificant trend for a higher expression of *B3gnt5* ($n = 6$ mice (three females and three males) per group; age, 8 months; *$P < 0.05$, Mann–Whitney test).

C–F   Confirming lower expression of the "A1" marker C3d, Western blot analysis indicated lower protein levels of C3d in APP/PS1-Stat3KO. Immunohistochemistry using an antibody against C3d revealed that lower expression of C3d particularly occurred in peri-plaque reactive astrocytes (arrows indicate C3d and GFAP colocalization; arrowheads indicate plaques visualized with methoxy-XO4; scale bars, 50 μm; APP/PS1-Stat3WT, $n = 6$ (two females and four males) mice; APP/PS1-Stat3KO, $n = 6$ (three females and three males) mice; age, 8 months; *$P < 0.05$, Mann–Whitney test).

G   Whole-brain levels of the pro-inflammatory cytokines IL-1β and TNF-α were significantly reduced in APP/PS1-Stat3KO mice (Mann–Whitney test), whereas no changes were seen for IL-10 (APP/PS1-Stat3WT, $n = 13$ (six females and seven males) mice; APP/PS1-Stat3KO, $n = 13$ (eight females and five males) mice; age, 11 months; *$P < 0.05$, Mann–Whitney test).

H   These changes were paralleled by a decrease in the area covered by dystrophic neurites in APP/PS1-Stat3KO compared to APP/PS1-Stat3WT mice, as assessed by LAMP1 staining (*$P < 0.05$, Mann–Whitney test; $n = 10$ male mice for both groups; age, 8 months; scale bars, 300 μm).

Data information: Data are represented as mean ± SEM.

(Fig 7C). Moreover, in the probe trial test performed 24 h after the last training day, APP/PS1-Stat3KO mice—similar to WT-Stat3KO and WT-Stat3WT mice—spent significantly more time in the target quadrant, whereas APP/PS1-Stat3WT spent equal time in all quadrants (Fig 7D and E), indicating that the conditional Stat3 deletion preserved spatial long-term memory in APP/PS1 mice. Interestingly,

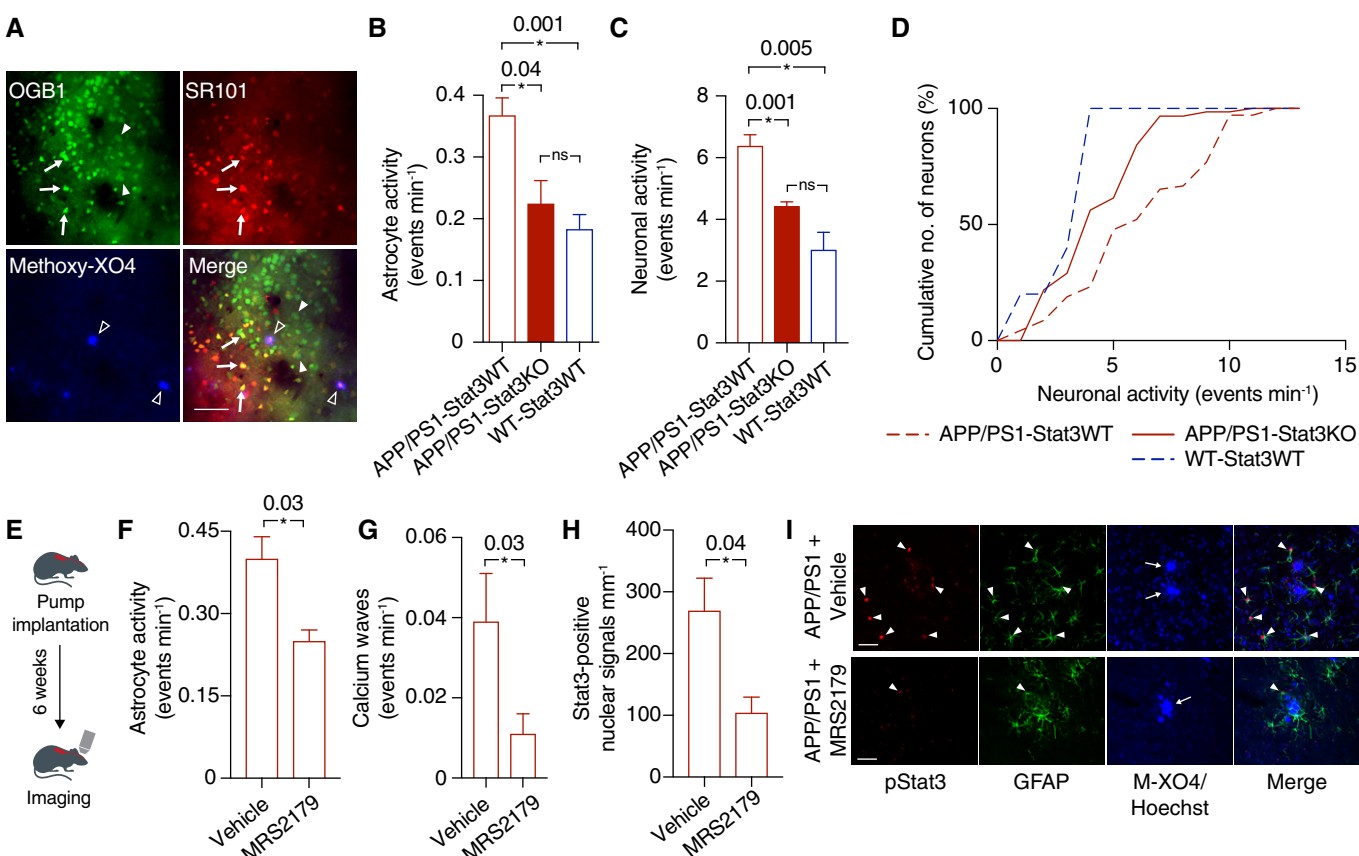

**Figure 6. Astroglial Stat3 modulates network imbalance in APP/PS1 mice.**

A    For *in vivo* two-photon imaging, astrocytes (arrows) and neurons (arrowheads) were labeled with the calcium indicator OGB-1, and astrocytes were co-labeled with sulforhodamine 101 (SR101; arrows). Aβ plaques were labeled with the intravital dye methoxy-XO4 (open arrowheads). Scale bar, 50 μm.

B    Calcium imaging of anesthetized animals showed that the hyperactivity of astrocytes in APP/PS1-Stat3KO mice was reduced to levels comparable to WT-Stat3WT mice, but significantly increased in APP/PS1-Stat3WT mice (*$P < 0.05$, one-way ANOVA followed by Bonferroni's multiple comparison test; $n = 6$ mice (three females and three males) for each group; age, 8 months).

C, D    Similarly, neuronal activity was also reduced to levels comparable to WT-Stat3WT mice in APP/PS1-Stat3KO mice, but significantly increased in APP/PS1-Stat3WT mice (*$P < 0.05$, one-way ANOVA followed by Bonferroni's multiple comparison test; same mice as in B). The cumulative distributions of neuronal calcium transients in APP/PS1-Stat3KO mice were not different from those of WT-Stat3WT mice ($P = 0.31$, Kolmogorov–Smirnov test), but significantly different from those of APP/PS1-Stat3WT mice ($P = 0.001$, Kolmogorov–Smirnov test).

E    To investigate a reciprocal scenario in which astroglial hyperactivity reduces Stat3 activation, we implanted osmotic minipumps into APP/PS1 mice and treated them with the network-normalizing P2Y1R inhibitor MRS2179 or vehicle for 6 weeks.

F–I    Two-photon imaging of these mice confirmed that astrocytic hyperactivity and propagating calcium waves in APP/PS1 mice were reduced by MRS2179 ($n = 6$ (four females and two males) mice) compared to vehicle-treated APP/PS1 mice ($n = 6$ (two females and four males) mice; age, 8 months; *$P < 0.05$, Mann–Whitney test). This network normalization induced a reduction in activated Stat3 (pStat3) in astrocytes assessed in fixed brain sections of the same mice (*$P < 0.05$, Mann–Whitney test). Scale bars, 50 μm. Arrowheads indicate pStat3 and GFAP colocalization, arrows indicate plaques visualized with methoxy-XO4 (M-XO4).

Data information: Data are represented as mean ± SEM.

we detected a similar effect of astroglial Stat3 deletion on cognition in an older cohort of mice, paralleled by reduced plaque load and size (Fig EV3), indicating that the protective effects of Stat3 deletion persist into later stages of pathology.

## A preclinical systemic Stat3 inhibitor confers protection from cognitive decline

Finally, as a preclinical proof-of-concept study and to rule out effects other than those related to Stat3 deletion in our genetic model, we tested the efficacy of SH-4-54, a small-molecule blood–brain barrier-permeable Stat3 inhibitor that shows accumulation to sufficient and

pharmacologically meaningful levels in the brain after systemic administration (Haftchenary *et al*, 2013). APP/PS1 mice were treated with SH-4-54 by intraperitoneal injection for 6 weeks and compared to age-matched APP/PS1 mice treated with vehicle. Importantly, as assessed in the Morris Water Maze test, treatment with the Stat3 inhibitor led to a significantly steeper learning curve of the latency to reach the platform (Fig 8A and B), whereas the swimming velocity remained unchanged (Fig 8C). Moreover, in the probe trial test 24 h after the last training day, transgenic mice treated with SH-4-54 spent more time in the target quadrant, whereas no difference was seen APP/PS1 mice treated with vehicle (Fig 8D). Therefore, the positive cognitive effects seen in the genetic

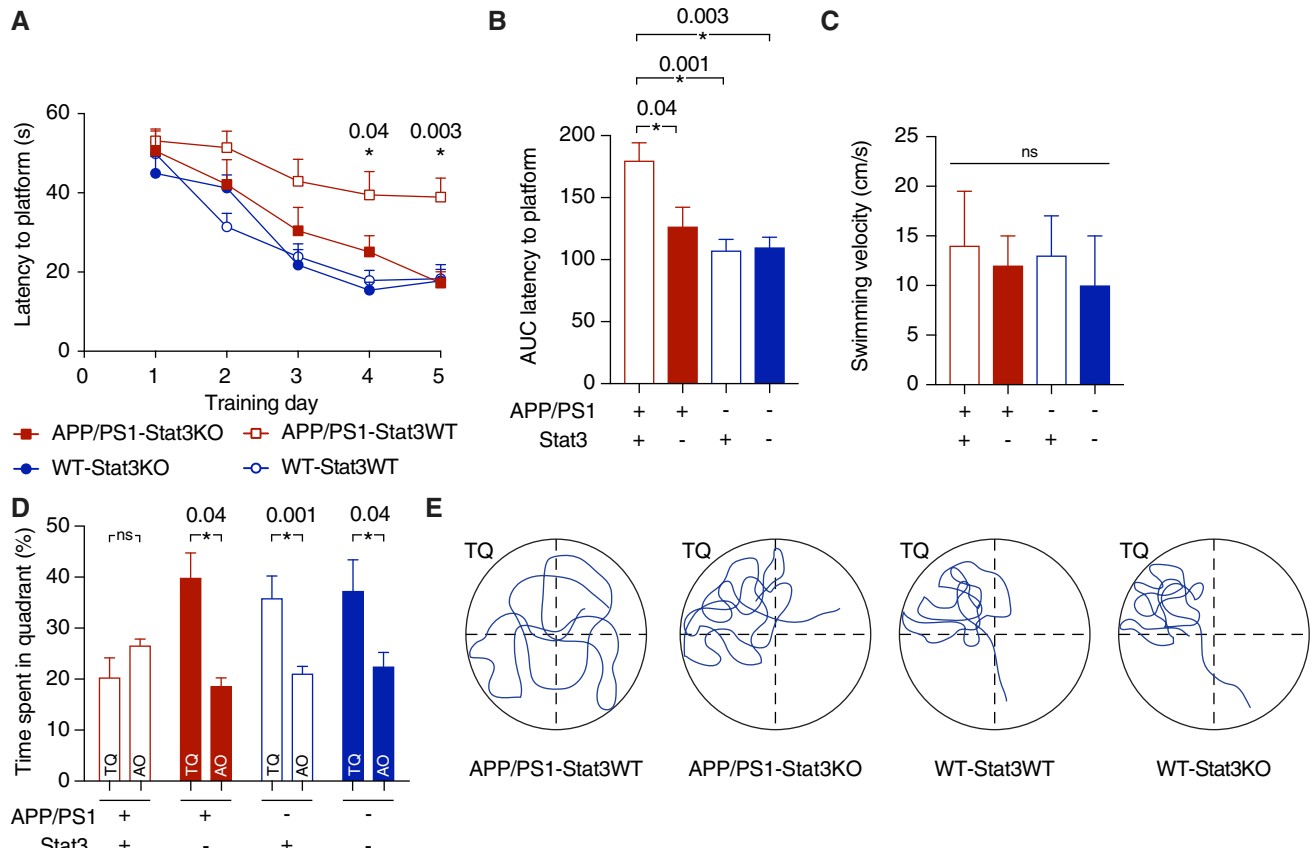

**Figure 7. Stat3 deletion in astrocytes protects from spatial memory and learning deterioration.**

A   Spatial learning and memory were assessed in the Morris Water Maze paradigm. APP/PS1-Stat3KO mice showed faster latencies to reach the hidden platform compared with APP/PS1-Stat3WT on days 4 and 5, but were similar to WT-Stat3WT and WT-Stat3KO mice (*$P < 0.05$, two-way repeated-measures ANOVA followed by Bonferroni post hoc test; $P$-values are for APP/PS1-Stat3KO versus APP/PS1-Stat3WT mice).

B   The area under the curve (AUC) for the latency to reach the hidden platform was similar in APP/PS1-Stat3WT compared to WT-Stat3WT and WT-Stat3KO mice, but significantly higher in APP/PS1-Stat3KO mice (*$P < 0.05$, Kruskal–Wallis test followed by Dunn's multiple comparisons test).

C   The swimming velocity was similar in all groups (Kruskal–Wallis test followed by Dunn's multiple comparisons test).

D, E   In the probe trial 24 h after the last training day, the time mice spent in the target quadrant (TQ) was different from chance in all groups except for APP/PS1-Stat3WT mice ($P < 0.05$, one-tailed one-sample $t$-test). APP/PS1-Stat3KO, WT-Stat3WT, and WT-Stat3KO mice spent significantly more time in the target quadrant compared to the mean of all other quadrants (AO), whereas APP/PS1-Stat3WT spent equal times in the target and all other quadrants (*$P < 0.05$, Wilcoxon matched-pairs signed rank test for each comparison).

Data information: Data are represented as mean ± SEM. WT-Stat3WT, $n = 15$ (7 females and 8 males) mice; WT-Stat3KO, $n = 15$ (10 females and 5 males) mice; APP/PS1-Stat3WT, $n = 15$ (eight females and seven males) mice; APP/PS1-Stat3KO, $n = 15$ (nine females and six males) mice; age, 8–9 months.

model were recapitulated by pharmacological inhibition in the AD model.

To confirm that SH-4-54 exerted meaningful biological effects in the brain in our paradigm, we subjected the same groups of mice after the behavioral phenotyping to *in vivo* two-photon imaging of cellular activity using the calcium indicator OGB-1. Similar to experiments in genetic Stat3KO mice, APP/PS1 mice treated with SH-4-54 showed an alleviation of neuronal hyperactivity compared to vehicle-treated APP/PS1 mice (Fig 8E).

As further confirmation, immunohistochemical analysis revealed that plaque size in APP/PS1 mice treated with SH-4-54 was slightly but significantly smaller compared to vehicle-treated mice, while plaque load and dystrophic neurite area remained unchanged (Fig 9A–D). Moreover, SH-4-54 treatment strongly reduced the fraction of pStat3-positive astrocytes (Fig 9E–G), indicating pharmaceutical

target engagement. Finally, similar to APP/PS1-Stat3KO mice, APP/PS1 mice treated with SH-4-54 displayed significantly longer process lengths of near-plaque microglia (Fig 9H–K).

## Discussion

In this study, we have shown that genetic modulation of astrogliosis confers protection from important pathological hallmarks and cognitive sequelae in a mouse model of AD. Specifically, we found that deleting Stat3 in the majority astrocytes induced a higher complexity of microglia around Aβ plaques, reduced amyloidosis and soluble Aβ, increased the internalization of Aβ by microglia, attenuated neuroinflammation, and normalized network homeostasis, ultimately resulting in lower dystrophic neurite burden and a better cognitive outcome.

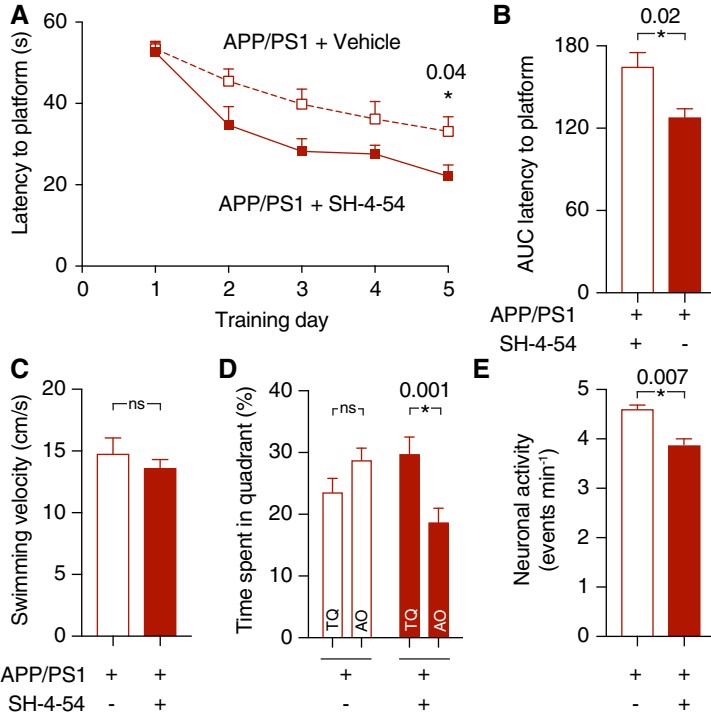

**Figure 8. The systemic Stat3 inhibitor SH-4-54 confers protection from cognitive decline in APP/PS1 mice.**

A   APP/PS1 mice were systemically treated with SH-4-54 for 6 weeks and compared to age-matched APP/PS1 mice treated with vehicle. In the Morris Water Maze test, treatment with SH-4-54 led to significantly faster latencies to reach the hidden platform on the last training day (*$P < 0.05$, two-way repeated-measures ANOVA followed by Bonferroni post hoc test).

B   The area under the curve (AUC) for the latency to reach the hidden platform was also smaller in APP/PS1 mice treated with the Stat3 inhibitor (*$P < 0.05$, Mann–Whitney test).

C   The swimming velocity was not affected by the treatment (Mann–Whitney test).

D   In the probe trial test, APP/PS1 mice treated with the Stat3 inhibitor spent significantly more time in the target quadrant (TQ) compared to the mean of all other quadrants (AO), whereas APP/PS1 mice treated with vehicle spent equal times in the target and all other quadrants (*$P < 0.05$, Wilcoxon matched-pairs signed rank test for all comparisons).

E   To verify target engagement in the brain, the same mice assessed in the Morris Water Maze were anesthetized and imaged using *in vivo* two-photon microscopy of calcium activity. Systemic treatment with the Stat3 inhibitor reduced the hyperactive phenotype of cortical neurons (*$P < 0.05$, Mann–Whitney test).

Data information: Data are represented as mean ± SEM. APP/PS1-Stat3WT, $n = 12$ (five females and seven males) mice; APP/PS1-Stat3KO, $n = 12$ (four females and eight males) mice; age, 8 months).

Importantly, the effects on cognition and network function were recapitulated by chronic treatment with a systemic Stat3 inhibitor.

Reactive astrogliosis has traditionally been considered a uniform response mechanism of the brain to acute or chronic injury. However, recent studies have challenged that notion by demonstrating that reactive astrocytes can create either detrimental or beneficial conditions for damaged neurons, likely depending on the nature, severity, and duration of the pathological stimulus (Zamanian *et al*, 2012; Liddelow *et al*, 2017). Whether protective ("A2") or neurotoxic ("A1") astrocytes represent truly distinct groups or two ends of a spectrum remains to be determined, but these data indicate that reactive astrocytes play key roles in orchestrating the multicellular response to neurodegeneration. Moreover, the above-mentioned studies have provided evidence that many neurotoxic pathways are directly executed by astrocytes, with microglia perhaps playing a triggering or initiating role (Liddelow *et al*, 2017; Rothhammer *et al*, 2018; Yun *et al*, 2018). Interestingly, we found that astroglial deletion of Stat3 resulted in reduced expression of some "A1" and increased expression of some "A2" transcripts in

APP/PS1 mice, perhaps indicating that Stat3 contributes to a context-dependent shift from toxic to protective astrocyte phenotypes. Moreover, we also found that C3d, a characteristic product of "A1" astrocytes (Liddelow *et al*, 2017) that contributes to AD pathology (Shi *et al*, 2017), is strongly reduced by astrocyte Stat3 deletion. However, our data also indicate that at least in the context of AD, the picture may be more complex. While many of the observed effects—such as cerebral network normalization—are indeed likely attributable primarily to astrocytes (Reichenbach *et al*, 2018), we also found evidence for the reciprocal scenario in which astrocytes may recruit microglia as downstream effector cells. For example, while we saw little overall phagocytosed Aβ in astrocytes and only minor morphological changes in astrocytes induced by the Stat3 deletion, microglial morphology was strongly altered to a more complex phenotype associated with a better containment of Aβ diffusion in previous studies (Condello *et al*, 2015). Hence, astroglial Stat3 signaling may mediate an astrocyte–microglia crosstalk that may "shield" the peri-plaque tissue from the diffusion of plaque-originating toxic Aβ species. Moreover, microglia incorporated

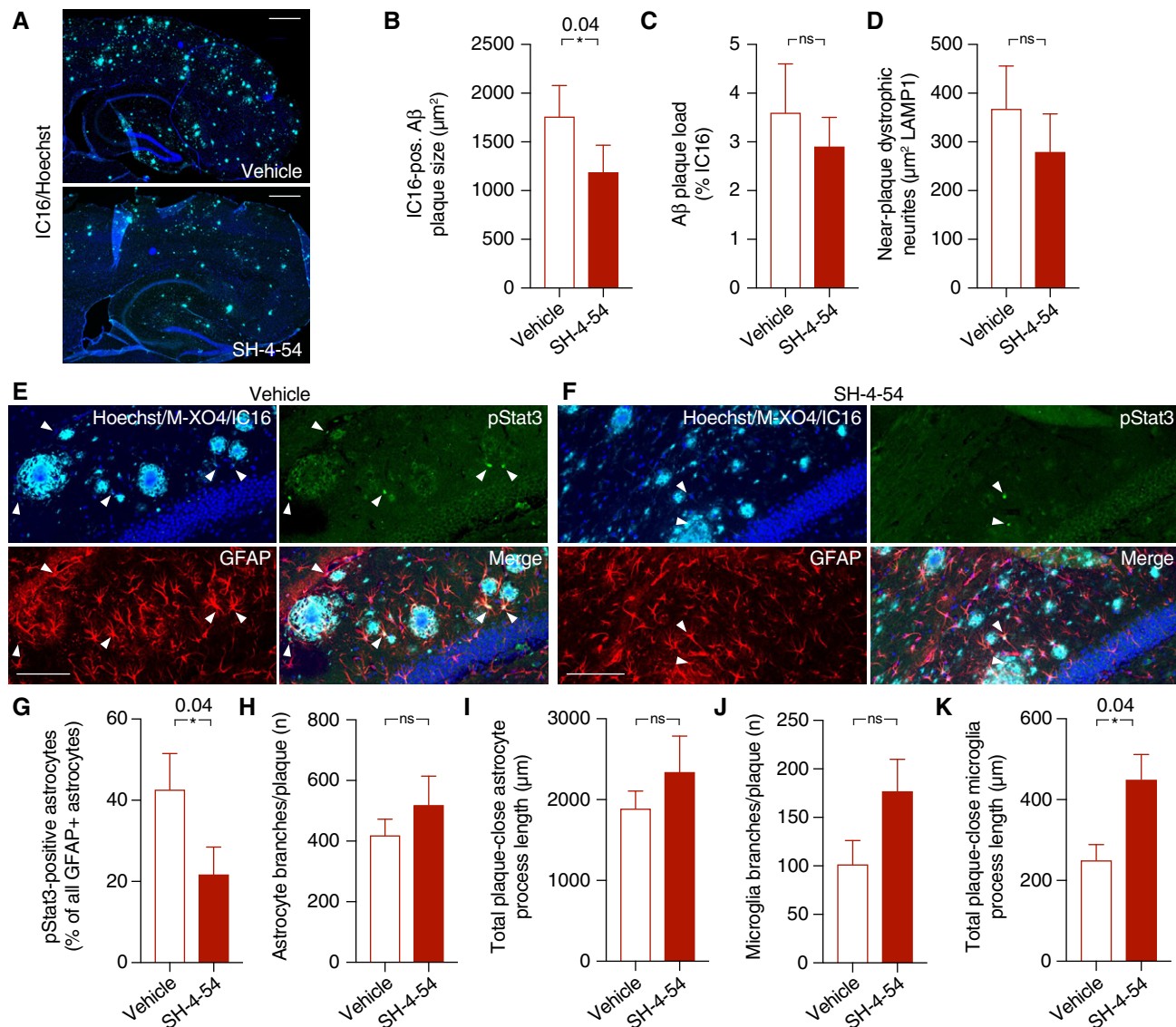

**Figure 9.  Target engagement of the systemic Stat3 inhibitor in APP/PS1 mice.**

A–D   SH-4-54 significantly decreased plaque growth, as assessed by IC16 immunohistochemistry, while plaque load remained unchanged. There was also no significant change in dystrophic neurite area during the treatment time (Mann–Whitney test for all comparisons; scale bars, 500 μm).

E–G   The fraction of pStat3-positive astrocytes in the peri-plaque region was strongly reduced by the treatment with the Stat3 inhibitor (arrowheads indicate pStat3 signals; scale bars, 100 μm; Mann–Whitney test).

H–K   While no changes were seen in morphological parameters of peri-plaque astrocytes, there was a significant increase in the process length of near-plaque microglia, indicating higher microglial complexity (Mann–Whitney test for all comparisons).

Data information: Data are represented as mean ± SEM. APP/PS1 mice treated with SH-4-54, $n = 12$ (six females and six males) mice; APP/PS1 mice treated with vehicle, $n = 12$ (7 females and 5 males) mice; age, 8 months.

significantly more Aβ, indicating—together with a changed expression of neprilysin, CD36, and apoE, and lower levels of pro-inflammatory microglia- or astrocyte-derived cytokines—an environment facilitating Aβ clearance. Hence, our data indicate that both microglia and astrocytes can reciprocally initiate and execute pathways relevant for neuronal demise or recovery. While astroglial Stat3 activation was strongly reduced in our model, we did not achieve full deletion of Stat3 from all astrocytes, similar to other models of Cre-mediated recombination. However, our data show that even a strong but only partial modulation of astrogliosis is sufficient for therapeutically relevant effects. Of note, in a recent study employing hippocampal virus-mediated overexpression of the Stat3 oppressor Socs3, altered genetic profiles and amyloid load were detected, but only marginal effects on cognition and phagocytosis (Ceyzeriat *et al*, 2018)[†].

[†]Correction added on 7 February 2019, after first online publication: Reference to Ceyzeriat *et al* was inserted.

While methodological differences such as hippocampal versus whole-brain effects may explain some of these differences, it is also interesting to note that Stat3 has Socs3-independent effects and vice versa (Panopoulos *et al*, 2006). In line with our results and attesting to the translational relevance, reduction in C3d receptor-mediated Stat3 signaling reduced neuroinflammation and pathology in a tau model of AD (Litvinchuk *et al*, 2018).

Importantly, Stat3 deletion not only affected the classical hallmarks of AD pathology, but also had strong positive effects on cerebral network dysfunction, that is, a clinically highly relevant target in AD (Palop & Mucke, 2016; Reichenbach *et al*, 2018). Interestingly, however, we here also found that network dysfunction itself may be a novel and potentially clinically relevant triggering factor of Stat3-mediated astrogliosis in AD. The pathways underlying this mechanism remain to be elucidated, but may be related to the fact that P2Y1 receptors—which are important mediators of network imbalance in AD (Delekate *et al*, 2014; Reichenbach *et al*, 2018)—as well as downstream astroglial calcium changes may activate astrogliosis and astrocytic Stat3 (Washburn & Neary, 2006; Kanemaru *et al*, 2013). The alternative scenario—that is, that P2Y1R inhibition decreases Stat3 activation and that the reduced calcium hyperactivity occurs downstream of this event—is less likely, as P2Y1R inhibition normalizes calcium activity in AD models within minutes (Delekate *et al*, 2014; Reichenbach *et al*, 2018).

Finally, the behavioral outcome of our genetic modulation was that APP/PS1 mice deficient for astroglial Stat3 showed a prominent preservation of spatial learning and memory at a relatively late disease stage. Hence, interference with these pathways holds the strong potential for clinically relevant and meaningful effects. Importantly, these positive behavioral effects were also seen in APP/PS1 mice treated with a systemically administrable, blood–brain barrier-permeable compound originally developed to target brain cancer (Haftchenary *et al*, 2013), indicating that available or novel drugs targeted at Stat3 or other mediators of astrogliosis are promising candidates for future preclinical and clinical AD trials.

# Materials and Methods

### Animals

All applicable international, national, and institutional guidelines for the care and use of animals were followed, and all experiments were approved by the Landesamt für Natur, Umwelt und Verbraucherschutz of North Rhine—Westphalia (Germany). The mice were on a C57BL/6N background. We used mice that co-express the human Mo/HuAPP695swe mutation in the amyloid precursor protein (APPswe) and human presenilin 1 (PS1-dE9; K595N/M596L) (APP/PS1) (Jankowsky *et al*, 2004) and their wild-type age-matched and sex-matched littermates. These mice were crossed to a line expressing tamoxifen-inducible CreERT under the Cx43 promoter (Eckardt *et al*, 2004) and with a line in which exons 12-14 of the Stat3 gene are flanked by loxP sites (Alonzi *et al*, 2001). Cx43-CreERT mice were also crossed with the Ai9 reporter line expressing a loxP-flanked STOP cassette upstream of the tdTomato sequence (B6.Cg-Gt(ROSA)26Sor^{tm9(CAG-tdTomato)Hze}/J). Animals were housed in groups on a 12-h light/dark cycle with food and water available

*ad libitum*. The age and sex of the animals in each experiment are provided in the figure legends.

### Pharmacology

Tamoxifen (5 μl/g; Sigma; solubilized in ethanol with sunflower oil) was injected i.p. in 6-week-old mice for 5 days. SH-4-54 (10 mg/kg; Merck) was solubilized in dimethyl sulfoxide (DMSO) and saline, and injected i.p. for 4 days followed by three injection-free days for 6 weeks. DMSO/saline was used as vehicle. MRS2179 (1 mM; Tocris) was solubilized in saline and applied intracerebroventricularly for 6 weeks (saline was used as a vehicle).

### Intracerebroventricular cannulation and pump implantation

Surgical implantation and cannulation were performed as described previously (Reichenbach *et al*, 2018). Briefly, mice were implanted s.c. at the interscapular region with osmotic minipumps (model Alzet 2006; delivery rate, 0.15 μl/h for 42 days; Durect) connected to Alzet Brain Infusion Kit 3 (anteroposterior [AP] −0.2 mm, medial lateral [ML] +1 mm relative to bregma; dorsal ventral [DV] +2.5 mm from the brain surface; Durect) for intracerebroventricular drug delivery. Animals were anesthetized with isoflurane (1.5% vol/vol) and O₂ (1 l/min) and kept on a heating pad (37°C) during surgery. Buprenorphine was used as an analgesic. Pumps were filled according to the manufacturer's instructions.

### *In vivo* two-photon microscopy

Cortical windows were implanted as described (Rakers & Petzold, 2017). Briefly, animals were anesthetized with isoflurane (induction, 3%; maintenance, 1–1.5% v/v in O₂) and kept on a heating pad (37°C). The scalp was removed after fixation in a stereotactic frame, and a craniotomy was created above the right somatosensory cortex using a dental drill. Agarose (1.5% in artificial cerebrospinal fluid) was placed on top of the cortex for stabilization, and the window was closed with a cover glass and sealed with dental cement.

OGB-1 AM (Life Technologies; solubilized in 20% Pluronic/80% DMSO and diluted to 1 mM with PBS) and SR101 (100 μM; Sigma) were co-injected into the cortex at a depth of 100–200 μm using a glass micropipette (tip diameter, 4–10 μm) connected to a pneumatic injector (1 bar, 60–90 s; PDES, NPI Electronic). Mice were anesthetized with isoflurane (0.5% v/v in O₂) and ketamine (50 mg/ml; 10 μl bolus followed by 1.5 μl/min, administered i.p. through a catheter connected to a pump) and kept on a heating pad (37°C). The mice were imaged using a Trim ScopeII microscope (LaVision) with three non-descanned detectors with band pass filters (620/60, 460/80, and 525/50 nm) or an LSM 7MP microscope (Zeiss) equipped with a Ti:sapphire laser (Chameleon Ultra II, Coherent) with three band pass filters (420/80, 500/550, and 565–610 nm) and one long pass filter (555 nm). The fluorophores were excited at 800 nm. A 20× W Plan-Apochromat (NA 1.0; Zeiss) was used for both microscopes. Z-stacks of the imaging region were taken (Trim ScopeII: XY, 250 × 250 μm, 828 px; Z, 1 μm stepsize, 100 μm range, pixel dwell time 0.93 μs; LSM 7MP: XY, 249.8 × 249.8 μm, 828 px; Z, 1 μm stepsize, 100 μm range, pixel dwell time 0.79 μs), followed by XY time-lapse series of calcium

activity (Trim ScopeII: 260 × 260 μm, 250 px, pixel dwell time 1.9 μs, frequency 3.55 Hz; LSM 7MP: 249 × 249 μm, 240 px, pixel dwell time 2.14 μs, frequency 3.47 Hz) for 10 min. Laser power below the objective was kept between 20 and 40 mW to minimize laser-induced artifacts and phototoxicity.

### Morris water maze assessment

Experiments were carried out using a circular pool (diameter, 110 cm) filled with opacified water at 20°C. The maze was virtually divided into four quadrants, with one containing a hidden platform (diameter, 10 cm) 1 cm below the water surface. Mice were placed in the water in a quasi-random fashion and were allowed to search for the platform for 60 s and remain on the platform for 15 s. If the mice did not reach the platform in the assigned time, they were placed onto it manually for 60 s. Extra-maze cues were present for spatial orientation. Mice were tested in four trials per day for five consecutive days with an inter-trial interval of 30 min. In the probe trial, the platform was removed and mice were put in a new position into the maze and allowed to swim for 60 s. All data were recorded and analyzed with EthoVision XT13 (Noldus).

### Immunohistochemistry

Mice were sacrificed and one hemisphere was fixed in 4% paraformaldehyde for 1 day, stored in sucrose (15 and 25%), and embedded in Tissue-Tek (Sakura). Sagittal sections (30 μm) were obtained using a cryostat (Thermo Fisher) and mounted onto slides. Postmortem human brain tissue sections (4 μm) from AD cases were obtained through a collaboration with the Department of Neuropathology (University Hospital Bonn, Germany) and deparaffined using xylene and descending ethanol solutions.

Sections stained for pStat3 were pre-treated for 45 min with 0.21% citric acid (90–95°C), followed by an incubation in 1% NaOH (w/v) and 2% $H_2O_2$ (w/v) for 20 min, 0.3% glycine (w/v) for 10 min, and 0.03% SDS (w/v) for 10 min at room temperature. All brain sections were blocked with 10% normal goat serum (Vector Labs) and 0.3% Triton X-100 (Sigma) in PBS for 1 h. Mouse brain sections were incubated with rat anti-GFAP (1:250; #130300; Invitrogen), rabbit anti-GFAP (1:500; #Z0334, Dako), mouse anti-Aβ (IC16; 1:250; provided by Dr. C. Pietrzik, Mainz University), rabbit anti-pStat3 (1:500; #9145S, Cell Signaling), rabbit anti-Iba1 (1:250; 019-19741, Wako), rabbit anti-Stat3 (1:500; #12640, Cell Signaling), rabbit anti-RFP (1:300; #AB234, Evrogen), rat anti-LAMP1 (1:750; #121602, BioLegend), mouse anti-S100β (1:250; #S2532, Sigma), rat anti-C3d (1:250; A0063, Dako), rabbit anti-Ki67 (1:500; RM9106, Thermo Scientific), mouse anti-NeuN (1:100; MAB377, Millipore), rabbit anti-NG2 (1:250; AB5320, Millipore), and rabbit anti-Olig2 (1:500; AB9610, Millipore) in 5% normal goat serum and 0.05% Triton X-100 overnight at 4°C. Plaques in human sections were stained with methoxy-XO4 (1:12,500, 1 h). Nuclei were stained with Hoechst 33258 (1:1,000; Thermo Fisher). Stainings were visualized with secondary antibodies (anti-rabbit Alexa 488, anti-rat Alexa 488, anti-mouse Alexa 633, anti-rat Alexa 633, anti-rabbit Alexa 633, anti-mouse Alexa 594, anti-rabbit Alexa 594; 1:1,000; Thermo Fisher). For thioflavin staining, sections were incubated with 70 and 80% ethanol and stained with 1% thioflavin S (Sigma) in 80% ethanol for 15 min. Afterward, sections were rinsed in 80% and

70% ethanol and distilled water. Images were acquired either using a confocal laser scanning microscope (LSM 700; Zeiss) with a 40× (NA 1.3) objective or a slide scanner (AxioScan.Z1, Zeiss) with a 10× objective (NA 0.45). The same image acquisition settings were used for each staining.

### Protein biochemistry

Mice were sacrificed, and one hemisphere was transferred to liquid nitrogen and stored at −80°C. Protein extraction was performed by homogenization in PBS (pH 7.4) with 1% phosphatase inhibitor and 1% protease inhibitor cocktails (Thermo Fisher) using a ceramic bead homogenizer (Precellys, VWR). Homogenates were extracted in RIPA buffer (in mM: 49.96 Tris, pH 7.2, 149.6 NaCl, 25.6 NP40, 24 NaDOC, 6.8 SDS) and centrifuged for 30 min at 100,000 g. The pellet containing insoluble Aβ was solubilized in SDS buffer (in mM: 69.2 SDS, 24.98 Tris, pH 7.5). Protein concentration was measured using a BCA Protein Assay Kit (Thermo Fisher) with a FLUOstar Omega Reader (BMG). Samples were solubilized in 4× LDS buffer (Life Technologies), boiled for 10 min at 95°C, centrifuged for 5 min at 20,000 g at 4°C, and loaded on 4–12% NuPAGE Novex Bis-Tris Midi-gels (Life Technologies) for protein gel electrophoresis. SeeBlue Plus2 pre-stained protein standard (Thermo Fisher) was used to determine molecular weights. Following electrophoresis, the samples were transferred to nitrocellulose membranes (0.2 μm; Bio-Rad). For TREM2 detection, samples were blotted on PVDF membranes. For CTF detection, the membranes were incubated for 1 min in 1× TBS (2.73 M NaCl; 39.9 mM Tris, 96°C, pH 7.6). Nitrocellulose membranes were blocked in 5% non-fat skimmed milk powder (Biomol) in 1× TBS with 0.05% Tween 20 (Merck), and PVDF membranes were blocked in 5% BSA in TBS for 1 h at room temperature. The following proteins were analyzed: full-length APP (6E10 antibody, 1:10,000; SIG-39320-1000, BioLegend), CTFs (C1/6.1 antibody, 1:1,000; 802801, BioLegend), neprilysin (anti-CD10, 1:500; AF1126, R&D Systems), apoE (anti-apoE, 1:500; AB947, Merck), TREM2 (anti-TREM2, 1:250; ab175525, Abcam), C3d (anti-C3d antibody, 1:200; A0063, Dako), CD36 (anti-CD36, 1:500; ab124515, Abcam), and actin (anti-β-actin, 1:8,000; A2103, Sigma). Immunoreactivity was detected by enhanced chemiluminescence reaction (Stella 3200, Raytest) using horseradish peroxidase-conjugated antibodies (anti-rabbit, 1:10,000, #7074, Cell Signaling; anti-mouse, 1:10,000, #115-035-062, Dianova; anti-goat/sheep, 1:3,000, A9452, Sigma) or by near-infrared detection (Odyssey, LI-COR) using IR Dye 800 CW antibody (1:20,000; #926-32211, LI-COR).

Quantification of Aβ and cytokines was performed using electrochemiluminescence multiplex ELISAs for $Aβ_{1–38}$, $Aβ_{1–40}$, and $Aβ_{1–42}$ [V-PLEX Aβ Peptide Panel 1 6E10 Kit, Meso Scale Discovery (MSD)] or cytokines (V-PLEX Pro-inflammatory Panel 1 Mouse Kit; MSD) according the manufacturer's instructions using a SECTOR Imager 2400 reader (MSD).

### Quantitative PCR

For total RNA extraction from dissected cortex, brain tissue was homogenized in QIAzol (Qiagen) and chloroform with a ceramic bead homogenizer (Precellys, VWR). RNA was purified using the RNeasy Plus Mini Kit (Qiagen), and cDNA synthesis was performed

## The paper explained

### Problem

The pathogenesis of Alzheimer's disease remains incompletely under-stood. Reactive astrogliosis, that is, activation of astrocytes, is an important hallmark of Alzheimer's disease, but the contribution of reactive astrocytes to pathological changes and cognitive decline associated with Alzheimer's disease remain largely unknown.

### Results

In a mouse model of Alzheimer's disease, we deleted Stat3, a canonical mediator of reactive astrogliosis, specifically in astrocytes. We found that this resulted in strongly attenuated pathological changes in the brains of these mice, along with better amyloid clearance and better cerebral network function, altogether resulting in a strong protection from cognitive decline. Importantly, these effects also occurred in Alzheimer's disease model mice treated with a pharmacological Stat3 inhibitor drug.

### Impact

Our data suggest that reactive astrogliosis, and specifically Stat3 activation in reactive astrocytes, is an important and pharmacologically addressable target in Alzheimer's disease.

using the high-capacity cDNA reverse transcription kit (Applied Biosystems) according to the manufacturer's instructions. Real-time PCR was carried out using the 7900 HT Fast Real-Time PCR System (Applied Biosystems) in 20 μl final volume, containing 10 μl of SYBRGreen PCR Master Mix (Applied Biosystems), 3 μl of a primer mix with a concentration of 1.5 μM of each primer, and 7 μl of cDNA (diluted 1:10). Primers ($5' > 3'$, forward; reverse) were used for *Actb* (ACCAGTTCGCCATGGATGAC; CTGAGAAAGTCAGAG TAGCTGA), *B3gnt5* (CGTGGGGCAATGAGAACTAT; CCCAGCT GAACTGAAGAAGG), *C3* (CCAGCTCCCCATTAGCTCTG; GCACTTG CCTCTTTAGGAAGTC), *Ggta1* (GTGAACAGCATGAGGGGTTT; GTT TTGTTGCCTCTGGGTGT), *Amigo2* (GAGGCGACCATAATGTCGTT; GCATCCAACAGTCCGATTCT), and *Tm4sf1* (GCCCAAGCATATTGT GGAGT; AGGGTAGGATGTGGCACAAG). Samples were analyzed in quadruplicates, and the expression levels of genes of interest were normalized to the expression of *Actb*.

### Data analysis

Cellular activity was analyzed as described (Reichenbach *et al*, 2018). Briefly, imaging data were imported into ImageJ and stabilized using the Image Stabilizer plugin. Regions of interest (ROIs) of OGB-1-positive cells were defined manually. OGB-1-positive cells were defined as astrocytes when they were co-labeled with SR101. Fluorescence over time (ΔF/F) was determined for each ROI and imported into Matlab R2013b (MathWorks). After removal of outliers using a median filter and smoothing using a Gaussian filter, individual signal parameters were determined using a custom-written algorithm in Matlab. Each time-lapse series that met this criterion was plotted together with the respective video file for visual inspection and verification.

All immunohistochemical data points represent mean values of 5–10 brain sections per mouse. Confocal images were imported into ImageJ, converted to 8-bit images, and smoothed using a Gaussian filter. After contrast enhancement (0.4% saturated

pixels), images were binarized using ImageJ (plaque stainings, automated MaxEntropy algorithm; GFAP and Iba1, a threshold defined as the mean background intensity plus the SD of background intensity multiplied by 2 was used). Astrocyte, microglia, and plaque area coverage as well as the number and size of plaques and LAMP1-positive dystrophic dendrites per area were quantified using ImageJ. The plaque area was subtracted from the area covered by dystrophic neurites before data analysis. Analysis of glial peri-plaque morphology was determined as described (Reichenbach *et al*, 2018). Briefly, following background removal, 3D particles were reconstructed using a Flood-Filler algorithm for each channel. Glial particles were considered as peri-plaque when they contained ≥ 1 voxels with a distance ≤ 2r to the plaque center. Near-plaque glial particles were smoothed using a Gaussian filter (Sigma, 1.0 px), skeletonized using the Skeletonize3D plugin, and analyzed using the Analyze Skeleton plugin. Astrocytes positive for pStat3 were defined as peri-plaque when they were located ≤ 50 μm around plaques.

To quantify Aβ engulfment by microglia and astrocytes, confocal z-stacks of mouse brain sections (*n* = 10 per animal) stained for Aβ (IC16 antibody and methoxy-XO4), GFAP, and Iba1 were background-subtracted and smoothed using ImageJ. 3D volume surface renderings of each *z*-stack were created using Imaris 8.4.1 (Bitplane), and the volume of glial cells and Aβ compartments (IC16 and methoxy-XO4) was determined from surface-rendered images. To quantify the volume of engulfed Aβ, fluorescence outside of astrocytes or microglia was subtracted from the image, and the remaining engulfed fluorescence was surface-rendered and calculated (Schafer *et al*, 2012, 2014).

### Statistical analysis

Sample size was predetermined based on a statistical power of 0.8 using G*Power 3 analysis software (Faul *et al*, 2007) and based on previous experience. Sample sizes were not altered during the course of the study. Data were excluded from the analysis if an animal died during or between experiments, or if it did not display any meaningful attempts or motivation to search for the hidden platform in the behavioral assessment ("floating" behavior) (Vorhees & Williams, 2006). No outliers were excluded from the datasets. All mice were randomly assigned to experimental groups. All studies were performed by investigators blinded to treatment groups and sample identity. We used the Mann–Whitney test for comparisons between two groups, the Wilcoxon matched-pairs signed rank test to compare paired groups, and the Kruskal–Wallis test followed by Dunn's multiple comparisons test to compare several groups. We used the two-way repeated-measures ANOVA and Bonferroni post hoc test for multiple measurements in the same groups and the Kolmogorov–Smirnov test to compare cumulative distributions. Data were analyzed using Prism 7 (GraphPad) and are represented as mean ± SEM. $P < 0.05$ was accepted as statistically significant.

**Expanded View** for this article is available online.

### Acknowledgements

We thank Theresa Schulte, Jan Peter, Christoph Moehl, Stephanie Weber, and Jan N. Hansen (DZNE, Bonn, Germany) for technical help and Claus Pietrzik

(Mainz University, Germany) for providing the IC16 antibody. This work was supported by grants to GCP from the European Union (EU) Joint Program, Neurodegenerative Disease Research program (JPND; Horizon 2020 Framework Programme, grant agreement 643417/DACAPO-AD), the Alzheimer Forschung Initiative (AFI), and the German Center for Neurodegenerative Diseases (DZNE). AH is member of the Deutsche Forschungsgemeinschaft (DFG) (DFG)-funded Cluster of Excellence ImmunoSensation (EXC 1023).

## Author contributions

GCP conceptualized and supervised the study. AH provided material and provided data analysis algorithms. NR and GCP established the methodology, analyzed data, and wrote the manuscript with input from all other authors. NR, AD, MP, FS, SK, and NB carried out the experiments and analyzed data.

## Conflict of interest

The authors declare that they have no conflict of interest.

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
