## [Review Process File · EMBO Molecular Medicine]

Inhibition of Stat3-mediated astroglial pathology ameliorates Alzheimer's disease model

Nicole Reichenbach, Andrea Delekate, Monika Plescher, Franziska Schmitt, Sybille Krauss, Nelli Blank, Annett Halle and Gabor C. Petzold

Review timeline:

Submission date:	9 August 2018
Editorial Decision:	12 September 2018
Revision received:	22 November 2018
Editorial Decision:	3 December 2018
Revision received:	4 December 2018
Accepted:	11 December 2018

Editor: Céline Carret

Transaction Report:

1st Editorial Decision

12 September 2018

Thank you for the submission of your manuscript to EMBO Molecular Medicine. We have now heard back from the three referees whom we asked to evaluate your manuscript.

You will see from the set of comments pasted below that the referees are supportive, still several questions need to be answered, details and clarification provided and additional experiments performed. In particular some mechanism would be required to better understand the downstream effect and devise an informed translational application. Further, a better documentation of the therapeutic effect is equally needed.

We would welcome the submission of a revised version within three months for further consideration and would like to encourage you to address all the criticisms raised as suggested to improve conclusiveness and clarity. Please note that EMBO Molecular Medicine strongly supports a single round of revision and that, as acceptance or rejection of the manuscript will depend on another round of review, your responses should be as complete as possible.

I look forward to receiving your revised manuscript.

***** Reviewer's comments *****

Referee #1 (Remarks for Author):

In this work Reichenbach and coworkers investigate the role of STAT3 signaling in astrocytes during AD. To this aim they generated an inducible conditional Ko mice driven by a Cx43-CreERT, backcrossed onto the APP/PS1 mouse model. In their studies the authors found a significant amelioration of the AD phenotype, as indicated by the decreased A-beta levels and plaque burden. Moreover, they detected an improvement in microglia phagocytic activity. These molecular findings were linked with an improvement in spatial learning and memory. These are interesting studies, properly controlled and described. However, it is my opinion that these studies fall short of providing a detailed understanding of the mechanisms through which STAT3 deletion in astrocytes ameliorates AD. Is this a direct effect? If that is the case, can the authors detect improved astrocyte phagocytic activity? Is it due to the increased phagocytic activity of microglia? What are the mechanisms involved?

Referee #2 (Remarks for Author):

Reichenbach and colleagues present a comprehensive study of the role of Stat3 signalling in reactive astrocytes in the context of mouse models of Alzheimer's disease (AD). Using the APP/PS1 amyloidosis model of AD, and extensive genetic ablation/silencing of largely astrocyte-specific Stat3 signalling, they report beneficial effects in memory and learning, along with changes in morphology of astrocytes and microglia, and an overall decrease in amyloid plaque load. With growing interest in the field of non-neuronal interactions in neurodegenerative diseases like AD, this study is likely to be of broad interest to the glia, degeneration, and broader neuroscience communities.

A few points would benefit from clarification:

1. does the APP/PS1 mouse have STAT3+ reactive astrocytes in the same location as human patients? Though some human post-mortem staining was provided (Fig 1G), it was unclear if this was also a peri-plaque region like that shown for mouse staining. This information is important for the reader to be able to ascertain the appropriateness of the mouse model.
2. Similarly, other groups have shown in recent years that STAT3-mediated reactive astrocytes are highly proliferative and produce a scar (in the context of acute injury) - was the same true in this mouse amyloidosis model?
3. What possible effects would the ~10% non-astrocyte specific targeting of the Cx43 mouse have on interpretation of these results? With around 5% of cells being non-astrocytes/non-neurons, if Stat3 signalling is sufficiently highly blocked in microglia this could account for the microglia-specific effects reported (e.g. increased phagocytosis of amyloid)
4. page 6, section titled 'Stat3 regulates plaque-associated...' the final sentence suggests that 'these data indicate that Stat3 signaling mediates an astrocyte-microglia crosstalk that may 'shield' the peri-plaque tissue...' there is no data for this conclusion, the conclusion to be drawn is that the astrocytes have an altered morphology in the peri-plaque region. Such prospective statements should not be included in the data section of the manuscript - please remove or move to the conclusions as a prediction to be further tested
5. Ca²⁺ imaging - did MRS2179 (P2Y₁R inhibitor) alter the individual spontaneous events, or where there changes in the network-wide propagating Ca²⁺ transients?
6. End of Ca²⁺ imaging section - the conclusion that Ca²⁺ transient changes can drive astrocyte reactivity has not been shown. MRS2179 can decrease Ca²⁺ transients AND decrease pSTAT3 immunofluorescence, but these data give no indication of a direct causative effect of calcium transients driving a reactive phenotype. Could it also not be that a decrease in Stat3 is driving Ca²⁺ changes? This conclusion is similarly easily drawn from these data and suggests that a decrease in STAT3 (ie. a decrease in reactivity) is driving calcium changes.

Overall this manuscript is well written, the data and figures are carefully prepared and easy to follow. I would imagine the study would be well-received by a broad readership. Aside from the few clarifications outlined above, I have no reservations about recommending this manuscript.

Referee #3 (Comments on Novelty/Model System for Author):

The authors used a complex way to generate their model and delete Stat3 in APPPS1 mice that does not confer full but partial Stat3 ablation. They should comment on this. Nonetheless, it is the first time Stat3 is deleted on a mouse model of AD and the novelty of the study is high. They also treated the mice with a Stat inhibitor so medical impact is potentially high. Technical quality is medium to high in most data of the MS.

Referee #3 (Remarks for Author):

By partially deleting Stat3 in astrocytes, Reichenbach et al. found decreased amyloid-beta load and neuritic dystrophy in the APPPS1 mouse model of AD, associated with several microglial alterations. Microglia become hypertrophic and increases phagocytosis and amyloid clearance pathways. A decrease of proinflammatory cytokines is also found. In addition, both Stat3 genetic deficiency and pharmacological inhibition decrease calcium hyperactivity in astrocytes and neurons and enhance learning and memory.

This work uses a variety of technical approaches to tackle an important and timely topic related to the astroglia-microglia crosstalk and the emerging essential role of these glial cells in the pathogenesis and progression of AD. However, deep revision needs to be done before being considered for publication in this journal.

1. The authors claim that the majority of reactive astrocytes in APPPS1-Stat3WT mice were Stat3+ while Stat3 activation was reduced by 80% in APPPS1-Stat3KO astrocytes, confirming its strong deletion.

Can the authors give exact numbers? Figure 1 shows that 50% of GFAP+ astrocytes are Stat3+ in the cortex and only 40% in hippocampus in APPPS1 mice and does not justify the sentence claiming that the majority of reactive astrocytes are Stat3+. Same applies for postmortem samples in which again only 40% of astrocytes are Stat3+. Moreover, around 10% of GFAP+ astrocytes are Stat3+ in both cortex and hippocampus in APPPS1-Stat3KO mice. This is neither an 80% reduction nor a strong deletion. I would also recommend adding a sentence discussing how such partial deletion of Stat3 in astrocytes leads to significant changes in AD pathology.

2. Figure 2: can authors show higher magnification images of astrocytes and microglia far from the plaques? Are there morphological differences between APPPS1-Stat3wt and KO in far areas? Is there any change in the number of cells?

3. Figure 4: the authors found a significant reduction of ApoE levels in APPPS1-Stat3 ko but it is not evident on the blots. Can they provide more representative images?

4. Whole brain levels of TNF α and IL1 β are reduced in APPPS1-Stat3 ko. Why the authors assume these proinflammatory cytokines are secreted by microglia? Astrocytes might contribute as well.

5. Stat3 pharmacological inhibition decreases hyperactivity and improved learning and memory. To which extent Stat3 was inhibited? Can the authors provide images and some quantifications? Is SH4-54 treatment having any effect on AB burden, neuritic dystrophy and astroglia and microglia morphology? Can they add data on these?

6. The authors do not mention in the text the time-points at which these characterizations were performed. Were the phenotypes more or less pronounced at different timepoints? Is there any variation over time?

Minor: there are very long sentences in the abstract and introduction that are difficult to understand. Can the authors split the information on separate sentences?

The images in Figure 3D seem to be upside down

1st Revision - authors' response

22 November 2018

Reviewer #1

In this work Reichenbach and coworkers investigate the role of STAT3 signaling in astrocytes

during AD. To this aim they generated an inducible conditional Ko mice driven by a Cx43-CreERT, backcrossed onto the APP/PS1 mouse model. In their studies the authors found a significant amelioration of the AD phenotype, as indicated by the decreased A-beta levels and plaque burden. Moreover, they detected an improvement in microglia phagocytic activity. These molecular findings were linked with an improvement in spatial learning and memory. These are interesting studies, properly controlled and described. However, it is my opinion that these studies fall short of providing a detailed understanding of the mechanisms through which STAT3 deletion in astrocytes ameliorates AD. Is this a direct effect? If that is the case, can the authors detect improved astrocyte phagocytic activity? Is it due to the increased phagocytic activity of microglia? What are the mechanisms involved?

RESPONSE: We thank the referee for the positive comments. We now provide the following new data strongly indicating that the underlying mechanism involves astrocytes directing microglia to increase their phagocytic capacity:

- We show in Figure 4 (reported on page 7) that deletion of astrocytic Stat3 increases the amount of A β phagocytosed by microglia, but not by astrocytes, indicating a mechanism that is initiated by astrocytes but executed by microglia.
- Along these lines, we now show in Figure 4 (and reported on page 7-8) that the microglia-specific A β -degrading proteins CD10/neprilysin and CD36 are strongly modulated by deletion of astrocytic Stat3, again indicating that the observed effects are mediated by modified astroglia acting on microglia. Similarly, ApoE expression was also reduced in APP/PS1Stat3KO mice. We have also examined TREM2 expression, but did not find major differences induced by Stat3 deletion.
- We now provide new qPCR and immunohistochemistry data, reported in the new Figure 5 and on page 8-9. These data show that deletion of astrocytic Stat3 reduces astroglial mRNA transcripts associated with the neurotoxic astrocytic phenotype termed 'A1' (Liddelow et al., Nature 2017), while increasing transcripts associated with the neuroprotective astrocytic 'A2' phenotype. We confirm these qPCR data by western blotting and immunohistochemistry against the important astroglial effector protein C3d, demonstrating that deletion of astrocytic Stat3 reduces the fraction of peri-plaque C3d-positive reactive astrocytes. Interestingly, this is in line with a very recent paper showing that C3-receptor deletion rescues tau pathology and attenuates neuroinflammation in a tau model of AD (Litvinchuk et al., Neuron 2018), and an earlier report that C3-deficient mice are protected from AD pathology (Shi et al., Sci Transl Med 2017). This is now discussed on pages 13 and 15.

Together, our data now strongly imply that the genetic modulation of reactive astrocytes, by inducing a phenotypical switch, directs microglia to increase their phagocytic capacity to better clear A β .

Reviewer #2

Reichenbach and colleagues present a comprehensive study of the role of Stat3 signalling in reactive astrocytes in the context of mouse models of Alzheimer's disease (AD). Using the APP/PS1 amyloidosis model of AD, and extensive genetic ablation/silencing of largely astrocyte-specific Stat3 signalling, they report beneficial effects in memory and learning, along with changes in morphology of astrocytes and microglia, and an overall decrease in amyloid plaque load. With growing interest in the field of non-neuronal interactions in neurodegenerative diseases like AD, this study is likely to be of broad interest to the glia, degeneration, and broader neuroscience communities. A few points would benefit from clarification:

1. does the APP/PS1 mouse have STAT3+ reactive astrocytes in the same location as human patients? Though some human post-mortem staining was provided (Fig 1G), it was unclear if this was also a peri-plaque region like that shown for mouse staining. This information is important for the reader to be able to ascertain the appropriateness of the mouse model.

RESPONSE: We now provide new stainings of human brain sections (shown and quantified in Figure 1 and described on page 6), which show that a significant number of pStat3-positive reactive astrocytes cluster around A β plaques (stained with methoxy-XO4) in human brain tissue from AD patients, very similar to what we have observed in APP/PS1 mouse brain. We now describe on page 23 that this analysis was specifically carried out in peri-plaque astrocytes.

2. Similarly, other groups have shown in recent years that STAT3-mediated reactive astrocytes are highly proliferative and produce a scar (in the context of acute injury) - was the same true in this mouse amyloidosis model?

RESPONSE: We have now tested this using immunohistochemistry against the cellular proliferation marker Ki67. Although this antibody was able to detect dividing/proliferating cells in the dentate gyrus as a positive control (now reported in the new Figure EV1), we detected few-to-none Ki67-positive (i.e. dividing/proliferating) reactive astrocytes around A β plaques in APP/PS1-Stat3WT or APP/PS1-Stat3KO mice (this is now reported on page 6 and in Figure EV1). This finding is in line with the current literature, given that reactive astrogliosis is a continuum that ranges from focal cellular hypertrophy to proliferation (i.e. scar formation; Sofroniew & Vinters, 2010), and that astrogliosis in Alzheimer's disease falls on the moderate end of the spectrum (Oberheim et al., J Neurosci 2008), with little-to-no astrocyte proliferation (Wang et al., Neurosci Bull 2018).

3. What possible effects would the ~10% non-astrocyte specific targeting of the Cx43 mouse have on interpretation of these results? With around 5% of cells being nonastrocytes/non-neurons, if Stat3 signalling is sufficiently highly blocked in microglia this could account for the microglia-specific effects reported (e.g. increased phagocytosis of amyloid)

RESPONSE: We have now better characterized the remaining 4.2 % (8 mo) and 6.2 % (11 mo) non-astrocytic non-neuronal cells using immunohistochemistry, and have found that these cells represent NG2 cells and a very small (<1 %) fraction of Olig2+ oligodendrocytes, but not microglia. This is now reported on page 5 and in Figure 1. Therefore, it is very unlikely that these effects were mediated by Stat3 deletion in microglia.

4. page 6, section titled 'Stat3 regulates plaque-associated...' the final sentence suggests that 'these data indicate that Stat3 signaling mediates an astrocyte-microglia crosstalk that may 'shield' the peri-plaque tissue...' there is no data for this conclusion, the conclusion to be drawn is that the astrocytes have an altered morphology in the periplaque region. Such prospective statements should not be included in the data section of the manuscript - please remove or move to the conclusions as a prediction to be further tested

RESPONSE: We agree, and have now moved this sentence to the Discussion. Other prospective statements and speculations were removed from the Results section as well.

5. Ca²⁺ imaging - did MRS2179 (P2Y₁R inhibitor) alter the individual spontaneous events, or where there changes in the network-wide propagating Ca²⁺ transients?

RESPONSE: This is a good point, as we have previously shown that MRS2179 also reduces the incidence of astroglial calcium waves (Delekate et al., 2014). We now provide data in Figure 6G (mentioned on page 10) that propagating astroglial calcium transients are reduced by P2Y₁R inhibition as well.

6. End of Ca²⁺ imaging section - the conclusion that Ca²⁺ transient changes can drive astrocyte reactivity has not been shown. MRS2179 can decrease Ca²⁺ transients AND decrease pSTAT3 immunofluorescence, but these data give no indication of a direct causative effect of calcium transients driving a reactive phenotype. Could it also not be that a decrease in Stat3 is driving Ca²⁺ changes? This conclusion is similarly easily drawn from these data and suggests that a decrease in STAT3 (ie. a decrease in reactivity) is driving calcium changes.

RESPONSE: We have now moved the discussion of this data to the Discussion. We agree with the referee that we have not directly shown a causative effect (this is now acknowledged on page 15). We now also discuss that a reverse sequence of events – a decrease in Stat3-mediated reactivity driving calcium changes – is also possible (page 15), but argue that this scenario may be less likely given that P2Y₁R inhibition normalizes calcium hyperactivity in AD models within minutes (as shown in our earlier papers: Delekate et al., 2014 and Reichenbach et al., 2018).

Overall this manuscript is well written, the data and figures are carefully prepared and easy to follow. I would imagine the study would be well-received by a broad readership. Aside from the few clarifications outlined above, I have no reservations about recommending this manuscript.

RESPONSE: We thank the referee for the positive comments.

Reviewer #3

Comments on Novelty/Model System for Author:

The authors used a complex way to generate their model and delete Stat3 in APPPS1 mice that does not confer full but partial Stat3 ablation. They should comment on this. Nonetheless, it is the first time Stat3 is deleted on a mouse model of AD and the novelty of the study is high. They also treated the mice with a Stat inhibitor so medical impact is potentially high. Technical quality is medium to high in most data of the MS.

RESPONSE: Thank you for the positive comments. We now mention on pages 5 and 13 that Stat3 was deleted “in the majority of astrocytes”. Moreover, we now discuss that our model does not confer full but partial ablation, but that this partial deletion is sufficient to achieve therapeutically relevant effects (page 14).

Remarks for Author:

By partially deleting Stat3 in astrocytes, Reichenbach et al. found decreased amyloid beta load and neuritic dystrophy in the APPPS1 mouse model of AD, associated with several microglial alterations. Microglia become hypertrophic and increases phagocytosis and amyloid clearance pathways. A decrease of proinflammatory cytokines is also found. In addition, both Stat3 genetic deficiency and pharmacological inhibition decrease calcium hyperactivity in astrocytes and neurons and enhance learning and memory.

This work uses a variety of technical approaches to tackle an important and timely topic related to the astroglia-microglia crosstalk and the emerging essential role of these glial cells in the pathogenesis and progression of AD. However, deep revision needs to be done before being considered for publication in this journal.

1. The authors claim that the majority of reactive astrocytes in APPPS1-Stat3WT mice were Stat3+ while Stat3 activation was reduced by 80% in APPPS1-Stat3KO astrocytes, confirming its strong deletion. Can the authors give exact numbers? Figure 1 shows that 50% of GFAP+ astrocytes are Stat3+ in the cortex and only 40% in hippocampus in APPPS1 mice and does not justify the sentence claiming that the majority of reactive astrocytes are Stat3+. Same applies for postmortem samples in which again only 40% of astrocytes are Stat3+. Moreover, around 10% of GFAP+ astrocytes are Stat3+ in both cortex and hippocampus in APPPS1-Stat3KO mice. This is neither an 80% reduction nor a strong deletion. I would also recommend adding a sentence discussing how such partial deletion of Stat3 in astrocytes leads to significant changes in AD pathology.

RESPONSE: In the original manuscript, we had indeed stated that the majority of reactive astrocytes in APP/PS1 mice were positive for Stat3. However, this statement was specifically a description of Stat3-positive astrocytes around plaques, which – as the images in Figure 1 show – is indeed the region where most of Stat3 immunoreactivity occurs. However, the graph in the original Figure 1 had reported the numbers for all astrocytes, regardless of plaque proximity, resulting in a discrepancy between what we reported in the text and what the graph showed. We now report the fraction of Stat3-positive astrocytes from all astrocytes in the text (page 5), and in addition report the number of peri-plaque astrocytes positive for Stat3 (page 5) in the new Figure 1. These data indeed confirm that the majority of peri-plaque astrocytes were Stat3-positive, and that this was reduced by ~80% in KO mice. Nevertheless, we agree with the referee that the deletion was only partial, and we now explicitly state this on page 14.

We also report the number of peri-plaque astrocytes positive for Stat3 in human sections in the new Figure 1, confirming that the majority of these astrocytes in human AD tissue was Stat3-positive.

2. Figure 2: can authors show higher magnification images of astrocytes and microglia far from the plaques? Are there morphological differences between APPPS1-Stat3wt and KO in far areas? Is there any change in the number of cells?

RESPONSE: We now provide higher-magnification images of astrocytes and microglia remote from

plaques in Figure EV2 (described on page 6). We also provide a quantification of morphological features and cell numbers in this figure, which shows no significant difference between the groups.

3. Figure 4: the authors found a significant reduction of ApoE levels in APPPS1-tat3 ko but it is not evident on the blots. Can they provide more representative images?

RESPONSE: We have repeated all Western Blot experiments, and now provide more representative images (Figure 4).

4. Whole brain levels of TNF α and IL1 β are reduced in APPPS1-Stat3 ko. Why the authors assume these proinflammatory cytokines are secreted by microglia? Astrocytes might contribute as well.

RESPONSE: Thank you for pointing this out. We have now changed the text to point out that microglia and astrocytes might both contribute to cytokine secretion (page 14).

5. Stat3 pharmacological inhibition decreases hyperactivity and improved learning and memory. To which extent Stat3 was inhibited? Can the authors provide images and some quantifications? Is SH4-54 treatment having any effect on AB burden, neuritic dystrophy and astroglia and microglia morphology? Can they add data on these?

RESPONSE: Heeding the referee's excellent point, we have performed new experiments now reported on page 12 and in the new Figure 9. Using immunohistochemistry, we now show that plaque size is significantly reduced after chronic treatment, while plaque load and dystrophic neurite area show nonsignificant trends towards a reduction, perhaps as expected given the relatively short treatment time. Moreover, we show that the relative number of pStat3-positive reactive astrocytes around plaques is significantly reduced in mice treated with SH-4-54 compared to controls. Finally, we find that total process length of near-plaque microglia is increased, similar to their morphology in APP/PS1-Stat3KO mice.

6. The authors do not mention in the text the time-points at which these characterizations were performed. Were the phenotypes more or less pronounced at different timepoints? Is there any variation over time?

RESPONSE: The time-points (i.e. age of the animals) are now reported in the Figure legends for all experiments and datasets. Most experiments were performed in mice aged 8-9 months old. Moreover, to investigate variation over time as requested by the referee, we now include data from 13-14 month-old mice, which is considered an advanced/late disease stage in the APP/PS1 model. These experiments, which are now reported in the new Figure EV3 and on page 10-11, show that the behavioral benefits, as well as reduced plaque load and size, persist at this later stage in APP/PS1-Stat3KO mice.

Minor: there are very long sentences in the abstract and introduction that are difficult to understand. Can the authors split the information on separate sentences?

RESPONSE: Thank you for pointing this out. This has now been corrected.

The images in Figure 3D seem to be upside down

RESPONSE: Thank you. We have corrected this.

Literature cited in reply to the referees' comments

Delekate A, Fächtemeier M, Schumacher T, Ulbrich C, Foddiss M, Petzold GC. Metabotropic P2Y1 receptor signalling mediates astrocytic hyperactivity in vivo in an Alzheimer's disease mouse model. *Nat Commun* 2014 5: 5422. doi: 10.1038/ncomms6422.

Liddel SA, Guttenplan KA, Clarke LE, Bennett FC, Bohlen CJ, Schirmer L, Bennett ML, Münch AE, Chung WS, Peterson TC, Wilton DK, Frouin A, Napier BA, Panicker N, Kumar M, Buckwalter MS, Rowitch DH, Dawson VL, Dawson TM, Stevens B, Barres BA. Neurotoxic reactive astrocytes are induced by activated microglia. *Nature* 2017 541: 481-487. doi: 10.1038/nature21029.

Litvinchuk A, Wan YW, Swartzlander DB, Chen F, Cole A, Propson NE, Wang Q, Zhang B, Liu Z, Zheng H. Complement C3aR Inactivation Attenuates Tau Pathology and Reverses an Immune Network Deregulated in Tauopathy Models and Alzheimer's Disease. *Neuron* 2018 doi: 10.1016/j.neuron.2018.10.031.

Oberheim NA, Tian GF, Han X, Peng W, Takano T, Ransom B, Nedergaard M. Loss of astrocytic domain organization in the epileptic brain. *J Neurosci* 2008 28: 3264-3276. doi: 10.1523/JNEUROSCI.4980-07.2008.

Reichenbach N, Delekate A, Breithausen B, Keppler K, Poll S, Schulte T, Peter J, Plescher M, Hansen JN, Blank N, Keller A, Fuhrmann M, Henneberger C, Halle A, Petzold GC. P2Y1 receptor blockade normalizes network dysfunction and cognition in an Alzheimer's disease model. *J Exp Med* 2018 215: 1649-1663. doi: 10.1084/jem.20171487.

Shi Q, Chowdhury S, Ma R, Le KX, Hong S, Caldarone BJ, Stevens B, Lemere CA. Complement C3 deficiency protects against neurodegeneration in aged plaque-rich APP/PS1 mice. *Sci Transl Med* 2017 9: eaaf6295. doi: 10.1126/scitranslmed.aaf6295.

Wang D, Zhang X, Wang M, Zhou D, Pan H, Shu Q, Sun B. Early Activation of Astrocytes does not Affect Amyloid Plaque Load in an Animal Model of Alzheimer's Disease. *Neurosci Bull* 2018 doi: 10.1007/s12264-018-0262-2.

2nd Editorial Decision

3 December 2018

Thank you for the submission of your revised manuscript to EMBO Molecular Medicine. We have now received the enclosed reports from the referees that were asked to re-assess it. As you will see the reviewers are now globally supportive and I am pleased to inform you that we will be able to accept your manuscript pending minor editorial amendments.

***** Reviewer's comments *****

Referee #1 (Remarks for Author):

The authors have addressed all my comments.

Referee #3 (Comments on Novelty/Model System for Author):

It is the first time Stat3 is deleted on a mouse model of AD and therefore the novelty is high. They also treated the mice with a Stat inhibitor so there is a potential medical impact. Technical quality is high in most data of the MS.

Referee #3 (Remarks for Author):

The authors answered all my questions. I consider that after revision the MS highly improved. It is of high interest and suitable for publication at EMBO molecular Medicine.

2nd Revision - authors' response

4 December 2018

Authors made the requested editorial changes.

Corresponding Author Name: Gabor C. Petzold

Manuscript Number: EM-2018-09665